# Animal Models of Pathogenic New World Arenaviruses

**DOI:** 10.3390/microorganisms13061358

**Published:** 2025-06-11

**Authors:** Alexander V. Alvarado, Robert W. Cross, Thomas W. Geisbert, Courtney Woolsey

**Affiliations:** 1Galveston National Laboratory, University of Texas Medical Branch, Galveston, TX 77555-0610, USA; alexalva@utmb.edu (A.V.A.); twgeisbe@utmb.edu (T.W.G.); 2Department of Microbiology and Immunology, University of Texas Medical Branch, Galveston, TX 77555-0610, USA

**Keywords:** arenaviruses, New World arenaviruses, Junín virus, Guanarito virus, Machupo virus, Chapare virus, Sabiá virus, animal models, countermeasures, pathogenesis

## Abstract

Since the emergence of Junín virus in 1953, pathogenic New World arenaviruses have remained a public health concern. These viruses, which also include Machupo virus, Guanarito virus, Sabiá virus, and Chapare virus, cause acute viral hemorrhagic fever and neurological complications, resulting in significant morbidity and mortality. Given the dearth of licensed therapeutics or vaccines against these pathogens, animal models of infection that recapitulate human manifestations of disease remain critically important to the development of efficacious medical countermeasures. Rodents and non-human primates have been successfully used to model human New World arenaviral infections, with guinea pigs, rhesus macaques, and cynomolgus macaques being the most successful models of infection for most major pathogenic New World arenaviruses. Here, we provide a highly comprehensive review of publicly reported animal models of pathogenic New World arenavirus infections, with a discussion of advantages and disadvantages for each model.

## 1. Introduction

Arenaviruses, or members of the family *Arenaviridae*, can be subdivided into New and Old-World complexes, depending on their region of endemicity [1]. New World arenaviruses (NWAVs) are restricted to the Western Hemisphere, whereas Old World arenaviruses (OWAVs) are endemic to the Eastern Hemisphere. Pathogenic NWAVs include Machupo virus (MACV, causative agent of Bolivian Hemorrhagic Fever (BHF), endemic to Bolivia), Guanarito virus (GTOV, causative agent of Venezuelan Hemorrhagic Fever (VHF), endemic to Venezuela), Chapare virus (CHAPV, causative agent of Chapare Hemorrhagic Fever (CHF), endemic to Bolivia), Sabiá virus (SABV, causative agent of Brazilian Hemorrhagic Fever, endemic to Brazil), Junín virus (JUNV, causative agent of Argentine Hemorrhagic Fever (AHF), endemic to Argentina), and Whitewater Arroyo virus (endemic to the United States). Pathogenic OWAVs include Lassa virus (LASV, the causative agent of Lassa fever, endemic to Nigeria, Sierra Leone, Liberia and Guinea; reported cases and/or infected rodent reservoirs in Togo, Mali, and Benin), Lujo virus (endemic to Zambia), and lymphocytic choriomeningitis virus (endemic globally) [1,2,3,4,5,6,7]. These pathogenic arenaviruses may lead to the development of acute viral hemorrhagic fever in humans [2,3,4,5,6,7,8]. Human infections largely stem from rodent–human interactions, including through exposure to excreta or fluids of infected rodents (via aerosolized particles, through consumption of contaminated food and/or water, or by direct rodent-human contact). Human-to-human transmission has also been reported, most commonly in nosocomial environments (as seen with LASV, MACV, and JUNV) [3,5,8]. The disease course is characterized by hemorrhagic and neurological complications following an initial course of non-specific febrile illness. Clinical manifestations, case counts, and mortality rates vary significantly among arenaviral diseases. Only sporadic cases of MACV and GTOV infections are reported annually, yet the mortality rates are high (i.e., 20–30%) [4,5]. By comparison, the case count and mortality rates associated with LASV infection are reportedly higher and lower, respectively, than those of NWAVs [3,9]. Nevertheless, these figures remain a subject of debate [1,3,4,5,9].

Regardless of the pathogenicity of arenaviruses or the nature of the diseases that some may cause in humans, all mammarenaviruses, New or Old-World, have bi-segmented, single-stranded, negative-sense RNA genomes (with each segment containing two genes) that utilize an ambisense coding strategy [10]. The glycoprotein precursor (GPC) and the nucleoprotein (N) are encoded on the small (S) segment, while the ring-finger matrix protein (Z) and the RNA-dependent RNA polymerase (L) are encoded on the large (L) segment [11,12]. The GPC plays a critical role in cell entry. Following RNA-dependent RNA polymerase-mediated mRNA synthesis and ribosomal translation, the GPC polyprotein is trafficked to the endoplasmic reticulum (ER) via a signal peptide [12,13,14]. Within the ER, this signal peptide is cleaved, yet it remains non-covalently associated with the glycoprotein, possibly stabilizing the entire complex [14]. Additional proteolytic processing of the GPC occurs in the Golgi complex, giving rise to two primary subunits: G1, which mediates receptor binding, and G2, which facilitates membrane fusion [14]. Once formed, the mature heterotrimeric glycoprotein complexes are trafficked to the host cell membrane, where they become anchored on the surface of nascent virions during budding [12,13,14].

For host cell entry, these glycoprotein spikes typically interact with alpha-dystroglycan (in certain OWAVs) or human transferrin receptor 1 (hTfR1, in most pathogenic NWAVs) via G1, leading to receptor-mediated endocytosis [12]. As the endosome matures and its pH lowers, the glycoprotein undergoes conformational changes that expose the G2 subunit, enabling fusion of the viral envelope with the endosomal membrane (a process which, rather uniquely in LASV, is facilitated by G1 switching from surface receptor alpha-dystroglycan to intracellular, endosomal receptor Lamp1) [15,16]. This fusion event delivers the viral genome into the cytoplasm, thereby initiating the infection process [12,13,14]. Most preclinical vaccines and therapeutics target arenavirus GPC and G1/2, or N to a lesser extent, as entry is a critical step in infection and future pathogenesis.

Currently, there are few medical countermeasures (MCMs) that are clinically available for the treatment or prevention of arenavirus diseases. Animal models are a critical component for MCM development, in addition to enabling researchers to better understand pathogenic mechanisms. This paper aims to discuss recent and past developments with regards to animal models for studying the five major pathogenic NWAVs: JUNV, MACV, GTOV, CHAPV, and SABV [1]. Further pertinent details regarding viral transmission, ecology, and clinical manifestations are also briefly discussed.

## 2. Junín Virus (JUNV)

### 2.1. Background

Junín virus is the causative agent of Argentine Hemorrhagic Fever (AHF) and was the first pathogenic NWAV to be characterized [8]. JUNV was first isolated in 1958, five years after the first cases of AHF were reported in 1953 [8,17]. Initial AHF cases occurred in rural, western, fertile plains of the Buenos Aires province near the city of Junín [8,17,18]. The zone of JUNV endemicity has considerably expanded across the pampas since the initial outbreaks to now include the southeast portion of the Córdoba province, the southern portion of the Santa Fe province, the northeast part of La Pampa province, and increasing swathes of the Buenos Aires province (with some recent cases of unknown etiology also found within the city of Buenos Aires) [17,18,19]. The natural reservoirs of JUNV are various small rodent species, primarily *Calomys musculinus*, but also *Calomys laucha*, *Akodon azarae*, *Bolomys obscurus*, and *Galictis cuja* [20,21]. *Akodon molinae* and *Calomys callidus* are capable of supporting chronic salivary viral shedding following experimental infection, but neither is confirmed as a wild carrier of JUNV [20]. Horizontal transmission is likely the primary means of transmission for rodents. Natural and experimental infections of newborn *C. musculinus* result in asymptomatic infections with viral shedding in urine and saliva as long as 480 days post-infection [22]. JUNV infection was also confirmed in the offspring of experimentally infected animals, pointing to the capacity for vertical transmission [22]. Human infection stems from rodent–human interactions, particularly during harvesting seasons, making agricultural workers in rural areas most at risk [8]. Exposure can occur via inhalation of aerosolized rodent excreta, contact at mucosal surfaces, or direct exposure through abraded skin [8,18]. In addition, human–human infections have been reported in nosocomial settings, presumably by direct exposure to the bodily fluids of an infected individual [8]. Human–human transmission may also occur through aerosolized particles or sexual transmission, but additional research is needed to confirm these potential routes [23,24,25]. Historically, hundreds of cases occur annually, but case numbers have significantly decreased in recent years, with fewer than 50 cases reported each year [18,26]. A decline in cases is partially attributed to the 2006 Argentine licensure of Candid#1, a live, attenuated JUNV vaccine consisting of a highly passaged sample of the XJ strain of JUNV [18,27]. Phase III clinical trials revealed 95% efficacy in protection from AHF, but a recurring risk of reversion to the more virulent strain prevents its wider adoption with regulatory agencies outside of Argentina [28]. Candid#1 use remains limited to non-pregnant, immunocompetent individuals over the age of 15 [27].

Phases of AHF infection and associated symptoms over time may be seen in Figure 1 below.

Kidney failure and renal dysfunction commonly occur at the end stage of disease [20,29]. Roughly 15–30% of all individuals will die as a result of infection, though mortality can be delayed through treatment with intravenous (i.v.) ribavirin (if provided early in the course of infection) or reduced through the infusion of convalescent plasma [18,29,30]. Following the acute phase, convalescence in survivors may last for a period of 1–3 months, characterized by memory and hair loss [29]. After an initial symptom-free period, a late neurological syndrome can develop 4–6 weeks post-symptom onset in approximately 10% of survivors receiving immune plasma. Late-onset neurological manifestations include fever and cerebellar signs, such as dysmetria, ataxia, intention tremor, slurred speech, or hypotonia [29,31]. Post-mortem pathological examinations reveal widespread necrosis of the liver and lymph nodes, pulmonary hemorrhage, interstitial pneumonia, alveolar damage, erythroblastopenia, and cerebral edema [8,18].

### 2.2. Animal Models of Experimental Infection with Junín Virus

Many preclinical animal models have helped characterize the pathogenesis of JUNV infection. These models have been instrumental in recapitulating human AHF clinical signs/pathologies and evaluating the efficacy of candidate treatments and vaccines. The most successful models include mice, rats, hamsters, guinea pigs, marmosets, capuchin monkeys, rhesus macaques, and cynomolgus macaques. Less successful models in replicating human AHF pathologies include howler monkeys, owl monkeys, three-striped night monkeys, pigs, and chickens [17,32,33,34]. Figure 2 compares common small animal models for JUNV, whereas Figure 3 compares common non-human primate (NHP) animal models for JUNV.

#### 2.2.1. Mice (*Mus musculus*)

Mice are widely used in JUNV research, with various strains and ages being studied. The first documented infection involved 1-day-old mice inoculated intracerebrally (i.c.) with JUNV XJ [17]. Clinical signs like encephalitis, tremors, and paralysis appeared within 9 days post-infection [17]. Adult mice were only susceptible to JUNV via intraperitoneal (i.p.) infection [17].

Subsequent studies found that JUNV infection in mice typically leads to the appearance of neurological clinical signs such as tremors, ataxia, and convulsions, usually starting 7–12 days post-infection, with natural death typically following 5 days after symptom onset [35]. Mortality rates are high (over 95%) in mice infected between 1 and 10 days old but decrease significantly as the mice age [35]. Histopathological lesions in the brains of infected newborn mice include vasculitis, perivasculitis, microglial activation, and neuronal degeneration [35].

Different mouse strains and routes of infection (i.c., i.p., subcutaneous [s.c.], intramuscular [i.m.]) have been tested [36,37,38]. Younger mice (1–10 days old) consistently showed high natural mortality and severe neurological clinical manifestations of disease [36]. Older mice showed reduced susceptibility and varied natural survival rates, with thymectomized mice demonstrating increased survival but persistent viral infection [36,38,39,40].

More recent studies using interferon-deficient and hTfR1-expressing mice revealed that younger mice are more susceptible to JUNV, with clinical signs like weight loss, lethargy, and neurological issues preceding humane endpoint euthanasia [41,42]. These studies confirmed that suckling mice are an effective model for studying the neurological effects of JUNV, though they are less effective at replicating the hemorrhagic clinical signs seen in humans. Despite some limitations, suckling and transgenic mice are valuable models for testing JUNV countermeasures due to their neurological manifestations and high mortality at low viral doses. Moreover, ample immunological reagents are available for mice compared to other small animal models such as guinea pigs and hamsters. A summary of reported experimental JUNV infections in mice of varying ages and strains may be seen below in Table 1.

#### 2.2.2. Table 1: Experimental JUNV Infection of Mice (*Mus musculus*)

**Table 1 microorganisms-13-01358-t001:** Table of studies of experimental JUNV infection of mice, displaying age of mice used, animal strain (if applicable), viral strain and passage history, viral inoculum size and route of infection, reported clinical manifestations of infection, average time to death, mortality rate, and source.

Age, Animal Strain (Noted If Applicable)	Virus Strain, Passage History, Inoculum, Route of Infection	Clinical Disease	Time-to-Death (Days Post-Infection)	Mortality Rate (%)	Source
Suckling white mice, aged 1 day	XJ strain (prototype), passaged twice or more in suckling guinea pigs, unknown viral titer i.c.	Encephalitis, incoordination with gait, tremors, convulsions, terminal hind-limb paralysis	Undisclosed	>0% (unspecified)	[17]
Adult white mice, age unspecified		Not reported	N/A	0% (unspecified)	
	XJ strain (prototype), passaged twice or more in suckling guinea pigs, unknown viral titer i.p.		Undisclosed	>0% (unspecified)	
Suckling Rockland, CFW, C3H, CF1, or Balb/c mice (outbred, immunocompetent), aged 1–10 days	Unspecified strain, passage history, and viral titer (known to be 10^3^ LD_50_ for 2-day-old mice only), i.c., s.c., and/or i.p.	Encephalitis, tremors, lateralized gait, convulsions, hind-limb paralysis	Within 5 days of symptom onset (12–17, estimated)	95–100% (unspecified), all routes and mouse strains/ages mentioned	[35]
Newborn, thymectomized Rockland, CFW, or Balb/c mice (outbred, immunosuppressed), unknown age		No reported clinical signs	Unspecified	~0% (unspecified)	
Juvenile Rockland, C3H, Balb/c, CFW and CF1 mice (outbred, immunocompetent), aged 15–30 days		No reported clinical signs	Unspecified	<95% (unspecified)	
Suckling CFW mice (outbred, immunocompetent), aged 1–30 days	XJ strain (prototype), passaged twice in guinea pigs and 13 times in suckling mouse brain, 5000 LD_50_ i.c.	Tremors, lateralized gait, convulsions, hind-limb paralysis	<14	100% (unspecified), 1, 3, 5, and 10-day-old mice	[36]
				85% (unspecified), 15 and 20-day-old mice	
				33% (unspecified), 25-day-old mice	
				7% (unspecified), 30-day-old mice	
	XJ strain (prototype), passaged twice in guinea pigs and 13 times in suckling mouse brain, 5000 LD_50_ i.p.	Tremors, lateralized gait, convulsions, hind-limb paralysis	<14	100% (unspecified), 1-day-old mice	
				16% (unspecified), 3-day-old mice	
				5% (unspecified), 6-day-old mice	
		No reported clinical signs	N/A	0% (unspecified), 9, 12, 15-day-old mice	
Newborn Rockland mice (outbred, immunocompetent), aged 1–2 days	RC strain, passaged at least once in suckling mouse brain, 1000 (suckling mouse) LD50 i.c.	Neurological manifestations of disease (not specified further)	12–17	100% (unspecified)	[39]
Newborn thymectomized Rockland mice (outbred, immunodeficient), aged 1–2 days		No reported clinical signs	Not reported	Near 0% (unspecified)	
Adult NIH pathogen free, nude mice with thymus (immunocompetent), aged 60 days	XJ strain (prototype), passaged 27 times in guinea pigs and 32 times in suckling mice, 1000 TCID_50_, i.c.	No clinical signs reported	Not reported	7.2% (4/55)	[40]
Adult NIH pathogen free, nude, thymectomized mice (immunosuppressed), aged 60 days			Not reported	3.6% (2/55)	
Adult C3H/HeJ (inbred, immunocompetent) mice, aged 45–180 days	XJ strain, passaged in suckling mouse brain (number of passages unclear), 1600 PFU i.c.	Tremor, ataxia, hyperkinesia	8.6	100% (unspecified)	[38]
	XJ strain, passaged in suckling mouse brain (number of passages unclear), 160 PFU i.c.	Excitability, hunched posture, hair standing, weight loss, fatigue, hypothermia, unresponsiveness, opistotonic neurological signs (3-month-old mice)	11.6	90% (unspecified)	
		Clinical signs not clearly indicated (45, 60, 120, 150, 180-day-old mice)	Not reported	>80% (unspecified), 45–120-day-old mice	
				~40% (unspecified), 150-day-old mice	
				~10% (unspecified), 180-day-old mice	
	XJ strain, passaged in suckling mouse brain (number of passages unclear), 160 PFU i.p.	No clinical signs reported	N/A	0% (unspecified), 3-month-old mice	
	XJ strain, passaged in suckling mouse brain (number of passages unclear), 160 PFU i.m.				
	XJ strain, passaged in suckling mouse brain (number of passages unclear), 160 PFU s.c.				
	XJ strain, passaged in suckling mouse brain (number of passages unclear), 16 PFU i.c.	Tremor, ataxia, hyperkinesia	10.3	100% (unspecified), 3-month-old mice	
	XJ strain, passaged in suckling mouse brain (number of passages unclear), 1.6 PFU i.c.		11	20% (unspecified), 3-month-old-mice	
	XJ strain, passaged in suckling mouse brain (number of passages unclear), 0.16 PFU i.c.	No clinical signs reported	N/A	0% (unspecified)	
Suckling C3H/HeJ (inbred, immunocompetent) mice, aged 1–15 days	XJ strain, passaged in suckling mouse brain (number of passages unclear), 160 PFU i.c.	Tremor, ataxia, hyperkinesia	Not reported	100% (unspecified), 1, 2, 3, 7-day-old mice	
				80% (unspecified), 15-day-old mice	
Adult C57BL, Balb/c, C3H/HeJ × BALB/cJ, and BALB/cJ × C3H/HeJ (inbred, immunocompetent) mice, aged 3 months		Not discussed (C57BL and Balb/c mice)	Not reported (C57BL and Balb/c mice)	10% (unspecified), C57BL and Balb/c mice	
		No clinical signs reported (C3H/HeJ × BALB/cJ, and BALB/cJ × C3H/HeJ mice)	N/A (C3H/HeJ × BALB/cJ, and BALB/cJ × C3H/HeJ mice)	0% (unspecified), C3H/HeJ × BALB/cJ, and BALB/cJ × C3H/HeJ mice	
Suckling albino (outbred) mice, aged 2–14 days	CbaFHA5069, passaged 5 times in suckling mice, unspecified viral titer i.c.	Not reported	Not reported	>0% (unspecified), all ages of mice	[37]
Adult IFN-α/β/γ R—/— mice (inbred, immune system suppressed) mice, aged 4–8 weeks	Romero strain, unclear passage history, 1 × 10^4^ PFU i.p.	Weight loss, scruffy coat, terminal decrease in body temperature	13.45	100% (13/13)	[41]
Adult Strain 129 twice backcrossed with C57BL/6 mice (in-bred, immune system competent) mice, aged 4–8 weeks		No clinical manifestations observed	N/A	0% (0/23)	
Adult hTfR1 HET mice, aged 3 weeks (inbred, immunocompetent)	Romero strain, passaged once in Vero cells, 10^5^ CCID_50_ (as measured in Vero cells), i.p.	Stagnation of weight gain, neurological signs of infection (e.g., unresponsiveness	14	>0% (1/unspecified)	[42]
Adult hTfR1 HOM mice, 3 weeks of age (inbred, immunocompetent	Romero strain, passaged once in Vero cells, 10^3^–10^5^ CCID_50_ (as measured in Vero cells), i.p.	Weight loss, lethargy, ruffling of fur, tremors, paralysis, abdominal distension, bleeding, encephalitis	13–16	100% (unspecified)	
			18		
			18	71% (5/7)	
Adult hTfR1 HOM mice, 4 weeks of age (inbred, immunocompetent)	Romero strain, passaged once in Vero cells, 10^5^ CCID_50_ (as measured in Vero cells), i.p.		13	16.6% (1/6)	
Adult hTfR1 HOM mice, 5 weeks of age (inbred, immunocompetent)		No clinical signs reported	N/A	0% (0/3 or 0/unspecified)	
Adult hTfR1 HOM mice, 6 weeks of age (inbred, immunocompetent					
Adult hybrid C57BL/6 × AG129 mice aged 3 weeks (immunosuppressed)					
Adult IFN- α/β R—/— mice, aged 3 weeks (inbred, immune system suppressed)	Romero strain, passaged once in Vero cells, 10^4^ CCID_50_ (as measured in Vero cells), i.p.				
Adult IFN- α/β/γ R—/— mice aged 3 weeks (inbred, immune system suppressed)					
Adult hTfR1 HOM IFN- α/β R—/— mice (inbred, immune system suppressed)	Romero strain, passaged once in Vero cells, 10^3^ CCID_50_ (as measured in Vero cells), i.p.				
Adult hTfR1 HOM IFN- α/β/γ R—/— mice aged 3 weeks (inbred, immune system suppressed)					
Newborn CFW mice (outbred, immunocompetent), aged 0–1 day	RC strain passaged in suckling mouse brain multiple times, unknown viral titer	Tremor, convulsions, paralysis	12–19	100% (50/50)	[43]
Newborn, thymectomized CFW mice (outbred, immunocompetent), aged 0–1 day		No reported clinical signs	N/A	0% (0/25)	

#### 2.2.3. Rats (*Rattus*)

Rats, both inbred and outbred, have been assessed as models of JUNV infection. In 1977, Wistar rats of various ages were infected i.c. with 1000 LD_50_ of JUNV XJ [44]. Symptomatic infection was observed in all rats aged 12 days or less, with natural mortality rates of >90% in 7–12-day-old rats [44]. Reported clinical signs included weight loss, diarrhea, conjunctivitis, tremors, convulsions, and neurological issues, especially in younger rats [44]. Rats aged 19 days or greater did not display clinical signs or experience JUNV-induced mortality [44].

In other studies, Buffalo/Sim inbred rats and outbred Wistar rats were infected i.p. or i.c. with varying doses of the XJ strain [45,46,47]. Mortality rates and the time to natural death varied with age, route, and dose, with younger rats largely showing higher susceptibility [46,47]. Neurological clinical signs like tremors, hyper-excitement, and paralysis were common [45,46,47].

Rats have also been used to model chronic JUNV virus infection [48]. In one study, 2-day-old Wistar rats were infected i.c. with 100,000 TCID_50_ of XJ [48]. The acute phase lasted 30 days, with observed clinical signs including inactivity, tremors, and hind-limb paresis [48]. After this phase, some rats displayed chronic clinical signs, such as tremors and gait abnormalities, over a 780-day period, though mortality was low [48].

These findings suggest that suckling rats, particularly Wistar and Buffalo/Sims strains, are useful for studying the neurological effects of JUNV and evaluating treatments. However, because JUNV is only lethal in very young rats, this model is not useful for evaluating vaccine efficacy. Further research is needed to explore the efficacy of rats as a model for other strains of JUNV beyond the XJ strain. A summary of reported experimental JUNV infections of rats of various ages and strains can be seen below in Table 2.

#### 2.2.4. Table 2: Experimental JUNV Infection of Rats (*Rattus*)

**Table 2 microorganisms-13-01358-t002:** Table of studies of experimental JUNV infection of rats, displaying age of mice used, animal strain (if applicable), viral strain and passage history, viral inoculum size and route of infection, reported clinical manifestations of infection, average time to death, mortality rate, and source.

Age, Animal Strain (Noted If Applicable)	Virus Strain, Passage History, Inoculum, Route of Infection	Clinical Disease	Time to Death (Days Post-Infection)	Mortality Rate (%)	Source
Suckling Wistar (outbred) rats, aged 2–3 days	XJ (prototype) strain, passaged in suckling mouse brain, 1000 LD_50_, i.c.	Weight loss, diarrhea, conjunctivitis, lateralization of gait, thinning of hair (and increased dullness of coat)	N/A	0% (unspecified)	[44]
Suckling Wistar (outbred) rats, aged 5 days			Not reported	31% (unspecified)	
Suckling Wistar (outbred) rats, aged 7 days				91% (unspecified)	
Suckling Wistar (outbred) rats, aged 10 days		Weight loss, diarrhea, conjunctivitis, lateralization of gait, thinning of hair (and increased dullness of coat), hyperexcitation, balance issues, cyanosis, tremors, and convulsions	12–13	93% (unspecified)	
Suckling Wistar (outbred) rats, aged 12 days		Weight loss, diarrhea, conjunctivitis, lateralization of gait, thinning of hair (and increased dullness of coat)	Not reported	91% (unspecified)	
Suckling Wistar (outbred) rats, aged 14 days				~30% (unspecified)	
Suckling Wistar (outbred) rats, aged 16 days				27% (unspecified)	
Juvenile Wistar (outbred) rats, aged 18 days				29% (unspecified)	
Juvenile Wistar (outbred) rats, aged 19 days		No clinical manifestations of infection reported	N/A	0% (unspecified)	
Juvenile Wistar (outbred) rats, aged 26 days					
Juvenile Wistar (outbred) rats, aged 28 days					
Juvenile Wistar (outbred) rats, aged 33 days					
Suckling rats, aged 2 days	XJ strain, passaged twice in guinea pigs and 15 times in hamsters, 10^3^ LD_50_, i.p.	Unspecified neurological signs	Not reported	85% (unspecified)	[45]
Suckling Wistar rats (outbred), aged 1 day	XJ strain (prototype), passaged in suckling mice brain, 10^3^ LD_50_, i.p.	Encephalitis; tremors; hyper-excitability; lateralized gait; hind-limb paralysis	19.9	69.2% (27/39)	[46]
Suckling Wistar rats (outbred), aged 2 days			20.49 (pooled average between three experiments)	85.3% (93/109) (pooled average between three experiments)	
Suckling Wistar rats (outbred), aged 2 days	XJ strain (prototype), passaged in suckling mice brain, 10^3^ LD_50_, i.c.	Not reported	Not reported	6.7% (1/15)	
Suckling Buffalo/Sim rats (inbred), aged 2 days				7.1% (1/14)	
Suckling Buffalo/Sim rats (inbred), aged 2 days	XJ strain (prototype), passaged in suckling mice brain, 10^3^ LD_50_, i.p.			88% (44/50)	
Suckling Wistar rats (outbred), aged 2 days	XJ strain (prototype), passaged in suckling mice brain, 10^2^ LD_50_, i.p.	Encephalitis; tremors; hyper-excitability; lateralized gait; hind-limb paralysis	24.11	50% (9/18)	
Suckling Wistar rats (outbred), aged 2 days	XJ strain (prototype), passaged in suckling mice brain, 10^4^ LD_50_, i.p.		18.7	70% (14/20)	
Suckling Wistar rats (outbred), aged 2 days	XJ strain (prototype), passaged in suckling mice brain, 10^5^ LD_50_, i.p.		19.25	63.2% (12/19)	
Suckling Wistar rats (outbred), aged 3 days	XJ strain (prototype), passaged in suckling mice brain, 10^3^ LD_50_, i.p.		22.22	34.6% (9/26)	
Suckling Wistar rats (outbred), aged 4 days			18	13.9% (5/36)	
Suckling Wistar rats (outbred), aged 5 days			23	12.9% (4/31)	
Suckling Wistar rats (outbred), aged 6 days		No clinical manifestations of infection reported	N/A	0% (0/31)	
Suckling Wistar rats (outbred), aged 7 days				0% (0/23)	
Suckling Wistar rats (outbred), aged 10 days				0% (0/56) (pooled data from two experiments)	
Suckling Buffalo/Sim rats (inbred), aged 10 days				0% (0/14)	
Suckling Wistar rats (outbred), aged 10 days	XJ strain (prototype), passaged in suckling mice brain, 10^3^ LD_50_, i.c.	Not reported	Not reported	95.2% (20/21)	
Suckling Buffalo/Sim rats (inbred), aged 10 days				81.3% (13/16)	
Suckling Wistar rats (outbred), aged 16 days	XJ strain (prototype), passaged in suckling mice brain, 10^3^ LD_50_, i.p.	No clinical manifestations of infection reported	N/A	0% (0/36) (pooled data from two experiments)	
Suckling Buffalo/Sim rats (inbred), aged 16 days				0% (0/8)	
Suckling Wistar rats (outbred), aged 16 days	XJ strain (prototype), passaged in suckling mice brain, 10^3^ LD_50_, i.c.			0% (0/12)	
Suckling Buffalo/Sim rats (inbred), aged 16 days				0% (0/10)	
Juvenile Wistar rats (outbred), aged 26 days	XJ strain (prototype), passaged in suckling mice brain, 10^3^ LD_50_, i.p.			0% (0/29) (pooled data from two experiments)	
Juvenile Wistar rats (outbred), aged 26 days	XJ strain (prototype), passaged in suckling mice brain, 10^3^ LD_50_, i.c.			0% (0/10)	
Juvenile Buffalo/Sim rats (inbred), aged 26 days				0% (0/12)	
Juvenile Buffalo/Sim rats (inbred), aged 26 days	XJ strain (prototype), passaged in suckling mice brain, 10^3^ LD_50_, i.p.			0% (0/13)	
Suckling Wistar (outbred) rats, aged 2 days	XJ strain (prototype), passaged at least once in suckling mouse brain, 100,000 Vero TCID_50_, i.c.	Up to 30 days post-infection: listlessness, tremors, hind-limb paresis and/or paralysis	Not reported	5% (unspecified)	[48]
		31–280 days post-infection: no clinical manifestations of disease reported		10% (unspecified)	
		281–780 days post-infection: tremors, lateralization of gait, hind-limb paralysis, blindness	N/A	0% (unspecified)	
Suckling Buffalo/Sim (inbred) rats, aged 8–12 days	XJ strain (prototype), passaged in suckling mice brain, 10^3^ PFU, i.c.	Neurological manifestations; encephalitis	Not reported	90–100% (unspecified)	[47]

#### 2.2.5. Hamsters (*Cricetinae*)

The effects of JUNV infection in hamsters have been studied across different ages and virus strains. Suckling hamsters (2–5 days old) were infected i.c. with dilutions of JUNV Cba Lye/63, Cba FHA 5054H, and Cba An 9446 [49]. While the clinical presentation of illness was broadly dose-dependent, exceptions were noted, such as with the Cba FHA 5054H strain, where all hamsters challenged with 1 (suckling mouse) LD_50_ got sick, while only 1 out of 5 hamsters challenged with 1000 LD_50_ became ill [49].

Hamster deaths typically occurred between 6 and 19 days post-infection, with mean time-to-death between 12 and 14 days post-infection, depending on the strain, though maternal cannibalism influenced recorded mortality times [49]. Similar results were observed in juvenile hamsters (7–19 days old), where neurological clinical signs included tremors, lack of coordination, and hind-limb paralysis, appearing around 10 days post-infection [49]. Some animals died or experienced developmental issues, but most survived JUNV infection [49].

In an earlier study, it was found that most infant hamsters (2 days old) infected i.p. with 1000 PFU of the XJ strain died naturally by day 12 post-infection [50]. Hematological analysis showed no significant changes in leukocyte or platelet levels, but a slight increase in glutamic oxalic transaminase was noted, suggesting possible liver dysfunction [50]. Collectively, these findings suggest that suckling hamsters could be a useful model for studying JUNV neuropathology and pathogenesis but are impractical for evaluating vaccine efficacy given their young age. A summary of reported experimental JUNV infections of hamsters of various ages can be seen below in Table 3.

#### 2.2.6. Table 3: Experimental JUNV Infection of Hamsters (*Cricetinae*)

**Table 3 microorganisms-13-01358-t003:** Table of studies of experimental JUNV infection of hamsters, displaying age of hamsters used, animal strain (if applicable), viral strain and passage history, viral inoculum size and route of infection, reported clinical manifestations of infection, average time to death, mortality rate, and source of study utilized.

Age, Animal Strain (Noted If Applicable)	Virus Strain, Passage History, Inoculum, Route of Infection	Clinical Disease	Time to Death (Days Post-Infection)	Mortality Rate (%)	Source
Suckling hamsters, aged 2–5 days	Cba Lye/63, passaged 8 times in suckling mice, dilutions of the LD_50_: LD_50_/100, LD_50_/10, LD_50_, 10 LD_50_, 100 LD_50_, 1000 LD_50_, 10,000 LD_50_ i.c.	Suckling hamsters: Unspecified	12.05 (suckling hamsters)	>0% (unspecified)	[49]
		Young hamsters: Lack of coordination in gait; excitability; hind-limb paralysis; prostration, underdevelopment	Unspecified (young hamsters)		
Juvenile hamsters, 7–19 days of age	Cba FHA 5045H, passaged twice in suckling hamster, dilutions of the LD_50_: LD_50_/100, LD_50_/10, LD_50_, 10 LD_50_, 100 LD_50_, 1000 LD_50_, 10,000 LD_50_ i.c.	Suckling hamsters: Unspecified	12.04 (suckling hamsters)		
		Young hamsters: Lack of coordination in gait; excitability; hind-limb paralysis; prostration	Unspecified (young hamsters)	50% (2/4) (young hamsters at each of 1000 or 10,000 LD_50_)	
	Cba An 9446, passaged 3 times in suckling mice, dilutions of the LD_50_: LD_50_/100, LD_50_/10, LD_50_, 10 LD_50_, 100 LD_50_, 1000 LD_50_, 10,000 LD_50_ i.c.	Suckling hamsters: Unspecified	13.82 (suckling hamsters)		
		Young hamsters: Lack of coordination in gait; excitability; hind-limb paralysis; prostration, underdevelopment	22 (young hamsters at 1000 or 10,000 LD_50_)		
Suckling hamsters, 2 days of age	XJ strain (prototype), unclear passage history, passaged many times in suckling guinea pigs and at least 17 times in suckling mice brains, 1000 PFU i.p.	Unspecified	12	>0% (unspecified)	[50]

#### 2.2.7. Guinea Pigs (*Cavia porcellus*)

Guinea pigs have been extensively studied as a model for JUNV infection across various strains. The XJ strain can cause symptomatic infection in outbred adult guinea pigs through various routes, including i.p., s.c., intranasally (i.n.), or orally [17,35,51]. Death typically follows naturally within 13–18 days post-infection, typically following emergence of hemorrhagic signs (associated with lesions such as spleen and bone marrow necrosis), dependent following infection with all routes except oral; uniform mortality can largely be obtained at higher doses regardless of route [17,23,35,51]. Reported clinical signs in guinea pigs include fever, weight loss, petechiae, intestinal congestion, terminal hypothermia, and cerebral congestion [35]. Hematological analysis revealed neutropenia, leukocytopenia, and thrombocytopenia as effects of JUNV XJ infection in outbred guinea pigs [52,53]. Various clotting factors (including II, V, VIII, IX, and XI) decreased over the course of infection, while quick time was lengthened, and partial thromboplastin time activated with kaolin was elevated [53]. Fibrin monomers were also detected in blood over days 7–13 post-infection [53].

Pregnant outbred guinea pigs infected with JUNV XJ experience high natural mortality rates within 9–15 days, with hemorrhagic clinical signs observed in both mothers and fetuses [54]. Viremia persists for up to 14 days post-infection in infected guinea pigs [55].

Intracerebral (i.c.) infection of outbred guinea pigs with CbaFHA5069 and CbaIV4454 strains results in moderate lethality, with time to natural death ranging from 10–26 days, accompanied by neuroinflammation, lymphocytic infiltrates, and paralysis [37]. Hind-limb paralysis and encephalitic lesions are hallmarks of neurological JUNV infection in guinea pigs [23]. Infection with the prototype strain through the same route yields weight loss and terminal wasting and hypothermia, but no clearly neurological clinical signs [56].

Other strains, such as Espindola, Romero, and Ledesma, also cause hemorrhagic manifestations and uniform mortality following infection of outbred guinea pigs, with average times-to-death by natural infection ranging from 14.5 to 19 days [57]. The P3551 strain induces both hemorrhagic and neurological clinical signs with an 80% mortality rate [57]. Infection with either of the Coronel or Suarez strain leads to similar neurological clinical manifestations but with lower mortality rates (10% and 40%, respectively) and extended time-to-death (24.5 to 30 days) [57].

Adult Hartley guinea pigs infected with 5 × 10^3^ LD_50_ i.m. of the Romero strain display fever, lymphocyte and granulocyte depletion, and significant weight loss prior to natural death to cause hemorrhaging [58,59]. Infection of younger Hartley guinea pigs with a more extensively passaged JUNV XJ strain at different possible doses (1 × 10^3^–1.5 × 10^3^ PFU i.p.) is associated with reduced mortality, whereas i.p. infection with Romero (7.5 × 10^3^ PFU) leads to uniform mortality (due to meeting humane endpoint criteria) [60]. Strain 13 guinea pigs exhibit similar strain-dependent responses, with i.p. Romero infection leading to similarly uniform lethality and clinical signs like encephalitis and hemorrhage, whereas XJ infection results in a milder course of disease [60].

Different JUNV strains elicit varying responses in Hartley guinea pigs [61]. For example, Espindola and Romero strains cause uniform mortality (in this case, death by natural infection), while Ledesma results in high but not uniform mortality [61]. Coronel and P3684 strains induce a neurological presentation, like paralysis with lower mortality rates, while P3551 results in a mix of hemorrhagic and neurological clinical signs with a higher mortality rate [61].

Given their high mortality rates and ability to replicate certain human hemorrhagic and neurological pathologies, guinea pigs are a valuable model for JUNV research. They have been used to test antiviral drugs, vaccine candidates, and other treatments, highlighting their versatility as an animal model for studying JUNV infection [23,27,37,62,63,64,65,66,67]. A summary of reported experimental JUNV infections of guinea pigs of various ages and strains can be seen below in Table 4.

#### 2.2.8. Table 4: Experimental JUNV Infection of Guinea Pigs (*Cavia porcellus*)

**Table 4 microorganisms-13-01358-t004:** Table of studies of experimental JUNV infection of guinea pigs, displaying age of guinea pigs used, animal strain (if applicable), viral strain and passage history, viral inoculum size and route of infection, reported clinical manifestations of infection, average time to death, mortality rate, and source.

Age, Animal Strain (Noted If Applicable)	Virus Strain, Passage History, Inoculum, Route of Infection	Clinical Disease	Time-to-Death (Days Post-Infection)	Mortality Rate (%)	Source
Unspecified age and strain	XJ strain (prototype), undescribed passage history, unknown viral titer i.p. or s.c.	Petechiae	12–15	100% (unspecified)	[17]
Adult, outbred guinea pigs, unspecified age	CbaFHA5069, passaged 5 times in suckling mouse brain, 0.80 log_10_ PFU i.c.	Unspecified	10–12	>0% (unspecified)	[37]
Suckling/juvenile outbred guinea pigs, aged 11 days	CbaIV4454, passaged 5 times in suckling mice, 0.70 log_10_ PFU i.c.	Hind-limb paralysis	10–26		
Unspecified age and strain	XJ strain (prototype), undescribed passage history, 100 LD_50_ s.c., i.p., i.m., i.n., i.c., or oral	Fever, weight loss, terminal hypothermia, petechiae	11–17	100% (unspecified)	[35]
Adult, pregnant guinea pigs (outbred), age unspecified	XJ strain (prototype), undescribed passage history, 10^3^ LD_50_ i.m.	Not reported	9–15	100% (5/5)	[54]
Adult Hartley guinea pigs (outbred), age unspecified)	XJ strain (prototype), undescribed passage history, 5 × 10^3^ LD_50_ i.m.	Initial fever, terminal hypothermia, petechiae	12.5	100% (10/10)	[59]
Adult Hartley guinea pigs (outbred), aged 1 year	Romero strain, passaged once in Vero cells, 7.5 × 10^3^ PFU i.p.	Shock, encephalitis, mucosal hemorrhage, coma, convulsions, paralysis	14–17	100% (4/4)	[60]
Juvenile Hartley guinea pigs (outbred), aged 5–10 weeks	Romero strain, passaged once in Vero cells, 1.5 × 10^3^, 2.5 × 10^3^, or 6.0 × 10^3^ PFU i.p.		9–19 (1.5 × 10^3^ PFU)	100% (3/3)	
			13–15 (2.5 × 10^3^ PFU)		
			12–17 (6.0 × 10^3^ PFU)	100% (4/4)	
	XJ strain (prototype), passaged 37 times in suckling mouse brain, passaged once in Vero cells, 1 × 10^3^–5 × 10^5^ PFU i.p.	Fever	N/A	0% (0/17)	
Juvenile strain 13 (inbred) guinea pigs, aged 8–20 weeks	XJ strain (prototype), passaged 37 times in suckling mouse brain, passaged once in Vero cells, 1 × 10^3^–1.5 × 10^3^ PFU i.p.			0% (0/9)	
Outbred guinea pigs, unspecified age	Espindola strain, passaged a low number of times in Vero cells, 10^3^ PFU, undisclosed route of infection	Primarily hemorrhagic manifestations of infection	17.3	100% (10/10)	[57]
	Ledesma strain, passaged a low number of times in Vero cells, 10^3^ PFU, undisclosed route of infection		19.0		
	Romero strain, passaged a low number of times in Vero cells, 10^3^ PFU, undisclosed route of infection		14.5		
	P3551 strain, passaged a low number of times in Vero cells, 10^3^ PFU, undisclosed route of infection	Mixed between hemorrhagic and neurological (non-suppurative encephalitis) clinical signs	21.1	80% (8/10)	
	Coronel strain, passaged a low number of times in Vero cells, 10^3^ PFU, undisclosed route of infection	Primarily neurological (non-suppurative encephalitis) clinical signs	30.0	10% (1/10)	
	Suarez strain, passaged a low number of times in Vero cells, 10^3^ PFU, undisclosed route of infection		24.5	40% (4/10)	
Adult Hartley (outbred) guinea pigs, age unspecified	Coronel (referred to as P3827 in pa-per), passaged twice in MRC-5 cells and once in Vero cells, 5000 PFU i.p.	Hind-limb paralysis	28	20% (4/20)	[61]
	P3551, passaged twice in MRC-5 cells and once in Vero cells, 5000 PFU i.p.		21.1	73.5% (11/15)	
	P3684, passaged twice in MRC-5 cells and once in Vero cells, 5000 PFU i.p.		27.8	40% (4/10)	
	Espindola (referred to as P3790 in paper), passaged twice in MRC-5 cells and once in Vero cells, 5000 PFU i.p.	Fatigue, weight loss, anorexia	17.3	100% (20/20)	
	Romero (referred to as P3235 in pa-per), passaged twice in MRC-5 cells and once in Vero cells, 5000 PFU i.p.		14.5		
	Ledesma (referred to as P3406 in paper), passaged twice in MRC-5 cells and once in Vero cells, 5000 PFU i.p.	Not specified	19	88.9% (16/18)	
Outbred guinea pigs, unspecified age	XJ strain (prototype), passaged twice in guinea pigs, 13 times in suckling mice, 25 additional times in guinea pigs and 19 additional times in suckling mice,300,000 or 30,000 TCID_50_ oral or 300,000, 30,000, or 3000 TCID_50_ i.n.	Not reported	Unspecified	100% (6/6)—300,000 or 30,000 TCID_50_ i.n.	[51]
				83% (5/6)—3000 TCID_50_ i.n.	
				40% (2/5)—300,000 TCID_50_ oral	
				60% (3/5)—30,000 TCID_50_ oral	
Unspecified age and strain	XJ strain (prototype), passaged at least once in guinea pigs, 10^7^ LD_50_ i.m.	Not reported	<20	>0% (unspecified)	[52]
Unspecified age and strain	XJ strain (prototype), passaged at least once in guinea pigs, 100 LD_50_ i.m.	Not reported	~14	100% (un-specified) * * Unknown proportion implied to have died naturally, most sacrificed during coagulation studies	[53]
Outbred guinea pigs, unspecified age	XJ strain (prototype), passaged at least once in mouse brain, 10^3^ PFU i.m.	Not reported	Not reported	Not reported	[55]
Outbred guinea pigs, unspecified age	XJ strain (prototype), passaged at least once in mouse brain, 10^3^ LD_50_ i.c.	Weight loss, terminal cachexia, hypothermia	11.3	100% (12/12)	[56]

#### 2.2.9. Common Marmosets (*Callithrix jacchus*)

Common marmosets are a well-characterized model of JUNV infection. Following intramuscular (i.m.) inoculation of marmoset adults with 1000 LD_50_ (as measured in guinea pigs) of the prototype XJ strain of JUNV, clinical signs emerge around 12–17 days, including general depression, anorexia, dehydration, adipsia, and weight loss [68]. These generalized clinical signs tend to increase in severity until death, which is preceded by terminal hypothermia [68,69,70,71]. Hemorrhagic and neurological signs include petechiae (observed on skin, and, following necropsy, observed on the adrenal glands), ecchymosis, erythematous rash, tremors, hyperexcitability, clonic spasms of the head and trunk, gingival hematomas, and tetanus-like convulsions [68,69,70,71]. Leukocytes, granulocytes, and erythrocytes can be detected in urine by 18 days post-infection, indicative of kidney dysfunction [68]. Hematomas and hemorrhages in the abdomen are commonly observed upon gross pathological examination [68]. Another study examining the effects of i.m. inoculation with the same variant and inoculum dose reported the presence of multifocal hemorrhages of the gums, pharynx, and esophagus, as well as occasional hemorrhaging in lymph nodes [69]. Meningoencephalitis, lymphoreticular perivascular cuffing, gliosis, and leptomeningitis have also been observed following microscopic examination of post-mortem brain tissue [69,70,71]. In the lung, interstitial pneumonia and a thickened alveolar septum were both observed, the former by day 14 post-infection, subsequently increasing in severity [69]. Hepatic necrosis (confluent or scattered), lymphocytic necrosis in the lymph node cortex, in splenic follicles and in the splenic red pulp, and focal necrosis in the bone marrow were also reported at various timepoints (after 14 days post-infection, in the case of the former two, and at 18- and 23- days post-infection for the latter) [69]. Leukocytopenia, thrombocytopenia, and anemia were noted following hematological analysis [69,70,71]. The high lethality and diversity of reported clinical signs recapitulated by marmosets have made it an attractive animal model for various applications. To date, marmosets have been used to evaluate the efficacy of Tacaribe virus as a vaccine for JUNV, as well as the efficacy of ribavirin and homologous immune sera as treatments for JUNV infections [72,73,74]. A summary of reported experimental JUNV infections of marmosets of various ages can be seen below in Table 5.

#### 2.2.10. Table 5: Experimental JUNV Infection of Common Marmosets (*Callithrix jacchus*)

**Table 5 microorganisms-13-01358-t005:** Table of described studies of experimental JUNV infection of common marmosets, displaying age of marmosets used, animal strain (if applicable), viral strain and passage history, viral inoculum size and route of infection, reported clinical manifestations of infection, average time to death, mortality rate, and source.

Age, Animal Strain (Noted If Applicable)	Virus Strain, Passage History, Inoculum, Route of Infection	Clinical Disease	Time-to-Death (Days Post-Infection)	Mortality Rate (%)	Source
Adults, age undisclosed	XJ strain (prototype), passaged in suckling mouse and guinea pig, 1000 LD_50_ (guinea pig) i.m.	General depression, anorexia, dehydration, adipsia, weight loss, petechiae, abdominal erythematous rash, ecchymosis, tremors, hyperexcitability, convulsions, terminal hypothermia, gingival hemorrhages and hematomas	22 * * Average of terminally moribund and natural death marmosets	66.7% (4/6) * * One died spontaneously, three were sacrificed when terminally moribund	[68]
	XJ strain (prototype), passaged 27 times in guinea pigs, 30 times in suckling mouse brain, 1000 LD_50_ (guinea pig) i.m.	Anorexia, weight loss, hyperexcitability, tremors, terminal hypothermia	23	10% (1/10)	[69]
	XJ strain (prototype), passage history unspecified, 1000 LD_50_ i.m.	Weight loss, general depression, anorexia, meningoencephalitis, tremor, post-stimulation clonic spasms of head and trunk, terminal hypothermia, unspecified hemorrhagic symptoms, gingival hematomas, difficulty walking		100% (2/2)	[70,71]

#### 2.2.11. Capuchin Monkeys (*Cebus* sp.)

Capuchin monkeys (genus *Cebus*) have been studied as a model for JUNV infection using various strains. I.M. infection of *Cebus apella* with 1.42 × 10^4^ PFU of JUNV Romero or 2.8 × 10^5^ PFU of JUNV P3551 resulted in mild clinical manifestations of infection, including anorexia and occasional temperature increases [58]. A slight neutrophil increase was noted in Romero-infected monkeys at day 10, with thrombocytopenia observed, though not severely enough to cause hemorrhaging [58]. No gross pathological lesions were found, though inflammatory infiltrates were present in the brain and CNS, along with slight neuronal necrosis in Romero-infected animals [58].

Another study involving four adult capuchins infected with JUNV XJ similarly reported mild clinical signs, including mouth congestion, gingivitis, polyadenopathy, and weight loss, which resolved by day 37 [75]. Transient leukocytopenia and thrombocytopenia occurred from day 7, peaking at day 14 before returning to normal [75]. One animal displayed neurological clinical signs (photophobia and tremors), which resolved by day 40 [75]. Viremia persisted between days 7–14, accompanied by high titers of neutralizing antibodies [75].

Together, these studies suggest JUNV infection in capuchin monkeys results in only mild disease, making them less ideal for evaluating the efficacy of MCMs. A summary of reported experimental JUNV infections of capuchin monkeys can be seen below in Table 6.

#### 2.2.12. Table 6: Experimental JUNV Infection of Capuchin Monkeys (*Cebus* sp.)

**Table 6 microorganisms-13-01358-t006:** Table of studies of experimental JUNV infection of capuchin monkeys/tufted capuchins, displaying age of monkeys used, animal strain (if applicable), viral strain and passage history, viral inoculum size and route of infection, reported clinical manifestations of infection, average time to death, mortality rate, and source.

Age, Animal Strain (Noted If Applicable)	Virus Strain, Passage History, Inoculum, Route of Infection	Clinical Disease	Time-to-Death (Days Post-Infection)	Mortality Rate (%)	Source
Adult *Cebus* sp., age unspecified	XJ strain (prototype), passaged 12 times in guinea pigs, passaged 13 times in suckling mice, and passaged an additional 27 times in guinea pigs, 10^4^ LD_50_ (as determined in guinea pigs) i.m.	Congested mouth, gingivitis, polyadenopathy, elevated body temperature, weight loss, photophobia, tremors	N/A	0% (0/4)	[75]
	P3551 strain, passaged twice in fetal rhesus macaque lung cells, passaged once in MRC-5 cells, 2.8 × 10^5^ PFU i.m.	Anorexia, possible mild lethargy and temperature increase			[58]
	Romero strain, passaged twice in MRC-5 cells, 1.42 × 10^4^ PFU i.m.	Anorexia, possible mild lethargy			

#### 2.2.13. Cynomolgus Macaques (*Macaca fascicularis*)

The use of cynomolgus macaques as a JUNV model has not been thoroughly characterized. In one study, infection of cynomolgus macaques with the XJ strain resulted in no reported clinical manifestations of infection [17]. In another study, adult macaques were infected i.v. with 5000 PFU of either JUNV Espindola or Romero [76]. All Espindola-infected macaques reached humane endpoints within 14–21 days and Romero-infected macaques within 13–21 days [76]. Clinical signs included significant weight loss, transient fever, facial edema, diarrhea, petechial rash, and neurological signs like ataxia and myoclonus [76]. Lymphocyte, monocyte, and granulocyte depletion, low platelet counts, elevated liver enzymes, and increased C-reactive protein were also observed [76].

Despite the limited number of reported studies, the high mortality rate and the similarity of observed clinical signs to AHF suggest that cynomolgus macaques could be a valuable model for studying JUNV pathogenesis and evaluating treatments and vaccines depending on the strain and/or route of challenge. Indeed, this model has been successfully used to evaluate the success of a chimeric monoclonal antibody treatment [76]. A summary of reported experimental JUNV infections of cynomolgus macaques can be seen below in Table 7.

#### 2.2.14. Table 7: Experimental JUNV Infection of Cynomolgus Macaques (*Macaca fascicularis*)

**Table 7 microorganisms-13-01358-t007:** Table of studies of experimental JUNV infection of cynomolgus macaques, displaying age of macaques used, animal strain (if applicable), viral strain and passage history, viral inoculum size and route of infection, reported clinical manifestations of infection, average time to death, mortality rate, and source.

Age, Animal Strain (Noted If Applicable)	Virus Strain, Passage History, Inoculum, and Route of Infection	Clinical Disease	Time-to-Death (Days Post-Infection)	Mortality Rate (%)	Source
Adult (3–6 years of age)	Romero strain, undisclosed passage history, 5000 PFU i.v.	Weight loss; transient fever; facial edema; diarrhea; petechial rash; weakness; ataxia; intention tremors; and seizures	18.7	100% (3/3)	[76]
	Espindola strain, undisclosed passage history, 5000 PFU i.v.		16.7		
Unspecified age	XJ strain (prototype), passaged twice in guinea pigs, unknown viral inoculum and route of infection	None reported	N/A	Not reported	[17]

#### 2.2.15. Rhesus Macaques (*Macaca mulatta*)

Infection of rhesus macaques with JUNV consistently results in high lethality and clinical manifestations resembling those observed in human AHF cases, regardless of strain. In one study, macaques were infected i.m. with 4.1–4.5 log_10_ Plaque Forming Units (PFU) of either the Espindola or Ledesma strain [77]. All animals infected with Espindola died naturally within an average of 33 days, displaying terminal dehydration, weight loss in excess of 25%, and hemorrhagic clinical signs like petechial rash and widespread mucosal membranous bleeding [23,77]. 71% of those infected with Ledesma died of natural infection within the same timeframe, exhibiting milder hemorrhagic clinical signs but more severe neurological clinical signs (i.e., tremors and ataxia) [77]. A later study also demonstrated the susceptibility of rhesus macaques to aerosol JUNV Espindola infection [78]. In total,

Leukocytopenia, lymphocytopenia, and granulocytopenia occurred transiently in rhesus macaques following JUNV infection, whereas platelet counts remained low until death [23,77]. Blood cultures revealed secondary bacterial infections leading to terminal bacteremia, specifically *Escherichia coli*, suggesting a potential association between JUNV pathology and secondary infections [77].

Post-mortem analysis of Espindola-infected animals revealed extensive pathological lesions, including spleen and bone marrow necrosis, hemorrhages in multiple organs, and pneumonia [23,79]. Ledesma-infected animals showed similar lesions but with lower incidence and additional neurological damage [79].

Histological examination indicated capillary congestion, neuronal degeneration, and lymphocytic infiltration, with more severe and earlier-onset neurological lesions in Ledesma infections compared to Espindola, aligning with findings in prior studies regarding their respective clinical courses of disease in rhesus macaques [57,79,80]. Other strains like P3551 and Romero showed varying mortality and disease manifestations in rhesus macaques, with P3551 causing a mix of hemorrhagic and neurological manifestations, and Romero leading to mild disease [57,80].

Rhesus macaques’ high mortality rates and similar pathology to human infections make them a valuable model for testing treatments or vaccines against JUNV [23,80,81]. They also exhibit strain-dependent variations in disease severity, mimicking human infections effectively [23]. Further investigation should assess the ability of Espindola-infected (but not Ledesma-infected) rhesus macaques to transmit virus to animals in adjacent cages when maintained in the same laminar flow isolators, to determine (1) possible aerosol transmission of virus between infected individuals in close proximity and (2) strain-specific differences in transmission [23]. A summary of reported experimental JUNV infections of rhesus macaques can be seen below in Table 8.

#### 2.2.16. Table 8: Experimental JUNV Infection of Rhesus Macaques (*Macaca mulatta*)

**Table 8 microorganisms-13-01358-t008:** Table of described studies of experimental JUNV infection of rhesus macaques, displaying age of macaques used, animal strain (if applicable), viral strain and passage history, viral inoculum size and route of infection, reported clinical manifestations of infection, average time to death, mortality rate, and source.

Age, Animal Strain (Noted If Applicable)	Virus Strain, Passage History, Inoculum, Route of Infection	Clinical Disease	Time-to-Death (Days Post-Infection)	Mortality Rate (%)	Source
Adults, age undisclosed	Espindola strain, passaged 3 times in MRC-5 cells, 4.1–4.5 log_10_ PFU, i.m.	Petechiae, ecchymoses, bleeding from mucosal membranes, terminal dehydration and weight loss	33	100% (8/8)	[23,77]
	Ledesma strain, passaged three times in MRC-5 cells, 4.1–4.5 log_10_ PFU, i.m.	Tremors, ataxia, paresis		71% (5/7)	
	Espindola strain, passaged low number of times in Vero cells, unspecified titer and route of infection	Hemorrhagic manifestations, no further specification provided	Not reported	100% (3/3)	[57]
	Ledesma strain, passaged low number of times in Vero cells, unspecified titer and route of infection	Neurological manifestations, no further specification provided			
	Romero strain, passaged low number of times in Vero cells, unspecified titer and route of infection	Mild clinical signs observed, no further specification provided	N/A	0% (0/3)	
	P3551 strain, passaged low number of times in Vero cells, unspecified titer and route of infection	Mixed neurological and hemorrhagic manifestations of infection, no further specification provided	Not reported	66.7% (2/3)	
	Romero strain, passaged 3 times in MRC-5 cells, unspecified titer and route of infection	Anorexia, fatigue, diarrhea or constipation, flushing of the face	N/A	0% (0/4)	[80]
	Espindola strain, passaged 3 times in MRC-5 cells, unspecified titer and route of infection	Progressive anorexia, malaise, diarrhea or constipation, facial erythema, malar or circumocular rash, conjunctivitis, oral ulcers, petechiae, gingival bleeding, serosanguinous nasal discharge, terminal hypothermia, dehydration and wasting, possible purulent conjunctivitis and oral ulcers	Not reported	100% (3/3)	
	Ledesma strain, passaged 3 times in MRC-5 cells, unspecified titer and route of infection			66.7% (2/3)	
	P3551 strain, passaged twice in rhesus fetal lung cells, unspecified titer and route of infection				
	Espindola strain, passaged twice in MRC-5 cells and once in Vero cells, 10^4^ PFU aerosolized	Anorexia, fatigue, weight loss, erythematous rash, gingival bleeding, gingival hemorrhage, bleeding from mucous membranes, wasting	29.5	100% (2/2)	[78]
	Espindola strain, passaged twice in MRC-5 cells and once in Vero cells, 10^2^ PFU aerosolized		31.3	100% (3/3)	

### 2.3. Summary of JUNV Animal Models

Guinea pigs, mice, hamsters, and rats each offer distinct benefits and limitations as small-animal models for AHF. Guinea pigs reliably reproduce hemorrhagic and some neurological manifestations, which—along with their relatively manageable size—makes them valuable for testing treatments and vaccines. However, they have fewer immunological reagents available compared to mice, limiting more advanced immune-response analyses. Mice are genetically well-characterized, offering abundant immunological tools and lower housing costs. Still, adult immunocompetent mice rarely exhibit hemorrhagic disease; lethality is typically confined to suckling or immunodeficient animals. Hamsters and rats, likewise, mainly develop serious or lethal outcomes when very young; in these age groups, some neurological symptoms and high mortality rates can be observed, but this age restriction diminishes their utility for vaccine efficacy evaluations.

NHP models—particularly rhesus macaques—most closely replicate the severe hemorrhagic and neurologic features of AHF. They consistently show high lethality and clinical symptoms resembling those observed in humans, making them excellent candidates for advanced pathogenesis studies and countermeasure testing. Cynomolgus macaques also demonstrate strain-dependent lethal infections, useful for exploring therapeutic interventions, although fewer overall studies have been performed. Common marmosets offer some of the same advantages as other NHP models but require specialized care and remain relatively costly and ethically challenging to use. Capuchin monkeys generally do not progress to severe disease and, thus, cannot fully mirror AHF’s hemorrhagic phenotype. Consequently, while rodent models are convenient and cost-effective for early research, the more resource-intensive NHP models provide the most faithful representation of human AHF.

## 3. Machupo Virus (MACV)

### 3.1. Background

The first recorded cases of Bolivian Hemorrhagic Fever (BHF) infection were reported in 1959–1964 (spread between two outbreaks, one from 1959–1962 and one from 1963–1964) [5,82]. These two initial outbreaks were concentrated in the Department of Beni in Bolivia, with cases occurring in the city of San Joaquín and the surrounding Llanos de Moxos region [5,82]. A total of 984 cases were recorded (470 in the first outbreak, 514 in the second outbreak), with 256 total deaths (142 from the first outbreak, 114 from the second outbreak), corresponding to a fatality rate of 26% [5]. Overall, BHF has a 15–30% case fatality rate, which is in line with these outbreaks [5]. Over the course of these outbreaks, the virus was isolated from a lethally infected patient’s spleen [83]. This specific strain was identified as Carvallo, the prototype MACV strain [83].

The reservoir for MACV is *Calomys callosus*, a rodent endemic to northern Bolivia. Experimental infection of *C. callosus* adults resulted in viruria and chronic viremia in 50% of animals for as long as 20 weeks post-infection [84,85]. Vertical transmission also seems to occur, as demonstrated through experimental infection of a pregnant *C. callosus* adult 10 days prior to delivery; all infants were viremic by 12 weeks of age [85]. Human contact with the excreta and/or secretions of infected animals, inhalation of aerosolized excreta/secretions, consumption of food contaminated with such excreta/secretions, or direct rodent–human contact can result in MACV infection [84]. While rodent–human exposure remains the most common route of MACV infection, human-to-human transmission has been sporadically reported. In 1971, a nosocomial outbreak occurred in the Cochabamba department of Bolivia, resulting in the deaths of 5 out of 6 total cases [86]. And in 1994, another episode erupted within a family after one member naturally contracted BHF [84].

BHF has been reported intermittently in Bolivia from 1959-present, with several outbreaks occurring since the initial 1959–1964 period, though case counts have never risen to the same level as were observed in the initial outbreaks—the largest subsequent outbreak was in 2007, with 20 cases and 3 deaths reported (within the Beni department); most other outbreaks have had single-digit case counts [5,87]. Most cases have been reported in the Beni department, though cases have also been reported in the Cochabamba, Santa Cruz and Tarija departments [87]. Clinical manifestations associated with MACV infection in humans can be seen in Figure 4 below.

### 3.2. Animal Models of Experimental Infection with Machupo Virus

Following its initial characterization in 1963, various strains of MACV have been used to experimentally infect animals to study viral pathogenesis and examine the efficacy of countermeasures. The best-characterized models include mice, hamsters, guinea pigs, Geoffroy’s tamarins, African green monkeys, rhesus macaques, and cynomolgus macaques. Animals that were less successful at replicating human BHF pathologies include house cats, horses, spiny rats, pigs, chickens, the common opossum, rodents of the genus *Oryzomys*, the Colombian white-faced capuchin, and the white-fronted capuchin [88]. The comparative benefits and downsides for the best-characterized rodent and NHP MACV models are displayed in Figure 5 and Figure 6. These animal models are described in greater detail in subsequent sub-sections.

#### 3.2.1. Mice (*Mus musculus*)

The response of various mouse strains to infection with MACV-Carvallo has been studied in detail. One report noted that i.c. infection of BALB/c mice resulted in uniform mortality within 8–9 days, likely due to an overactive immune response rather than organ damage, as no organ damage was observed post-mortem [89]. In contrast, adult C57BL/6 mice neither died nor displayed notable clinical signs post-infection [5,89,90]. Suckling C57BL/6 mice (aged 0–7 days) were more susceptible than adults, with deaths by natural infection reported after i.c. infection with 10^3^ hamster lethal doses (HLD_50_) of MACV [88]. Similar susceptibility was observed in other strains like AKR, DBA/2, C3H/HCN, and BALB/c mice [88].

Adult STAT-1 knockout mice have proven to be more effective models, showing high degrees of moribundity, leading to euthanasia, following infection with 1000 PFU MACV Carvallo via i.p. (100% mortality), i.n. (25% mortality), or s.c. (67% mortality) routes [91]. Mortality typically occurred within 7–11 days post-infection, with reported clinical signs including ruffled fur, hunched posture, and lethargy [91]. Pathological findings included thymic cortical atrophy, splenic lymphocyte death, peritonitis, necrotizing steatitis, and pancreatitis [91].

Suckling Swiss Webster mice infected i.c. with either of the Carvallo or Cochabamba strains (the latter isolated in a 1971 outbreak) exhibited growth retardation, tremors, convulsions, and death within 9–16 days [86]. Similar results were seen in Swiss Webster mice, with no significant difference in mortality from natural infection based on the route of infection (i.c., i.p.) [88,92].

Infection of thymectomized and non-thymectomized suckling Rockland mice (1–2 days old) with 1000 LD_50_ MACV-Carvallo (i.c.) resulted in uniform non-thymectomized mice mortality from natural infection within 12–17 days, displaying various histopathological brain lesions, whereas thymectomized mice survived the 40-day study duration with no significant lesions except lymphocyte depletion in those sacrificed at 30 or 40 days [39].

These findings suggest that suckling mice and certain immunosuppressed adult mice are effective models for studying the neurological effects of MACV but are less effective at replicating hemorrhagic pathologies. Certain mouse strains could be useful for countermeasure studies due to their high mortality rates under specific conditions. A summary of reported experimental MACV infections of mice of varying ages and strains can be seen below in Table 9.

#### 3.2.2. Table 9: Experimental MACV Infection of Mice (*Mus musculus*)

**Table 9 microorganisms-13-01358-t009:** Table of studies of experimental MACV infection of mice, displaying age of mice used, animal strain (if applicable), viral strain and passage history, viral inoculum size and route of infection, reported clinical manifestations of infection, average time to death, mortality rate, and source.

Age, Animal Strain (Noted If Applicable)	Virus Strain, Passage History, Inoculum, Route of Infection	Clinical Disease	Time to Death (Days Post-Infection)	Mortality Rate (%)	Source
Suckling NIH general purpose Swiss (outbred, immunocompetent) mice, aged 2 days or less	Carvallo (prototype), passaged eight or fewer times in suckling hamster or suckling mice, unknown viral titer of inoculum, i.c., i.p., or combined i.c./i.p.	Growth retardation; rough fur; dullness of fur coat; lack of coordination; tonic-clonic convulsions; ataxia; apnea; rigidity	Unspecified	>0% (unspecified)	[88,92]
Suckling BALB/c (inbred, immunocompetent) mice, aged 0–7 days	Carvallo (prototype), passaged eight or fewer times in suckling hamster or suckling mice, 10^3^ Hamster LD_50_ (for all other strains), i.c.				
Suckling C3H/HCN (inbred, immunocompetent) mice, aged 0–7 days					
Suckling AKR (inbred, immunocompetent) mice, aged 0–7 days					
Suckling DBA/2 (inbred, immunocompetent) mice, aged 0–7 days					
Suckling C57/6 (inbred, immunocompetent) mice, aged 0–7 days					
Adult NIH general purpose Swiss (outbred, immunocompetent) mice, aged 5 weeks or greater	Carvallo (prototype), passaged eight or fewer times in suckling hamster or suckling mice, unknown viral titer of inoculum, i.c. or i.p.	No clinical signs reported	Unspecified	0% (unspecified)	
BALB/c (inbred, competent immune system) mice of unspecified age	Unspecified strain and passage history, unspecified viral titer, i.c.	Unspecified	8–9	100% (unspecified)	[89]
C57B1/6 (inbred, competent immune system) mice of unspecified age			N/A	0% (unspecified)	
Suckling Swiss Webster Strain (outbred, competent immune system) mice, aged 3 days or less	Cochabamba, second passage from suckling hamster brain, 9 × 10^3^ PFU, i.c.	Growth retardation, tremors, convulsions	9–16	>0% (unspecified)	[86]
Suckling Swiss Webster Strain (outbred, competent immune system) mice, aged 3 days or less	Carvallo (prototype), second or third passage from suckling hamster brain, 2.1 × 10^3^ PFU, i.c.				
Adult STAT-1 knockout (inbred, immunosuppressed) mice, aged 6–12 weeks	Carvallo (prototype), passaged once in Vero cells, 1000 PFU, i.p.	Ruffled appearance, hunched posture, lethargy	7.3	100% (6/6)	[91]
	Carvallo (prototype), passaged once in Vero cells, 1000 PFU, i.n.		20	25% (1/4)	
	Carvallo (prototype), passaged once in Vero cells, 1000 PFU, s.c.		10.5	66.7% (4/6)	
IFN-α/β/γ R—/— mice (C5BL/6 background) (inbred, immuno-suppressed)	Carvallo (prototype), passaged once in Vero cells, 10,000 PFU, i.p.	Weight loss, neurological impairment (partial paralysis, hunched posture, labored breathing, awkward gait), hypothermia 1–2 days prior to death	22–34	84% (11/13)	[90]
C57BL/6 (inbred, competent immune system)		No clinical signs reported	N/A	0% (0/10)	
Newborn thymectomized Rockland mice (outbred, immunodeficient), aged 1–2 days	Carvallo (prototype) passaged at least once in suckling mouse brain, 1000 (suckling mouse) LD_50_ i.c.	No clinical signs reported	Unspecified	Near 0% (unspecified)	[39]
Newborn Rockland mice (outbred, immunodeficient), aged 1–2 days		Unspecified neurological clinical signs	12–17	100% (unspecified)	

#### 3.2.3. Hamsters (*Cricetinae*)

The response of hamsters to infection with MACV-Carvallo has been studied extensively. An initial study explored i.p. and i.c. routes [92]. Regardless of the infection route in hamsters younger than 5 days of age, clinical signs appeared within 7–18 days, including tonic-clonic convulsions, ataxia, apnea, and rigidity [92]. Mortality from natural infection varied greatly across litters [92]. Gross pathological lesions indicative of hemorrhagic disease were not observed, and older hamsters (4 weeks or older), regardless of route of infection, mostly survived [92].

Other studies found that the i.p. route was less effective in causing lethal and symptomatic infections, leading to asymptomatic infections in adult hamsters with chronic viral shedding [85,88]. Another study compared the pathogenesis of Carvallo with that of the Cochabamba strain in suckling Syrian golden hamsters [93]. Infection with Carvallo i.c. caused uniform mortality by day 11, with histopathological lesions including hepatic necrosis; lymphocytic depletion in the lymph nodes, thymus, and spleen; bone marrow hypoplasia; and pulmonary congestion [93]. No neurological signs were reported [93]. Cochabamba infection resulted in nonsuppurative encephalitis; reported lesions included perivascular cuffing, cerebellar necrosis, hepatic necrosis, and pulmonary congestion, with all hamsters naturally dying by day 17 [93].

These findings suggest that while suckling hamsters are a good model for studying the neurological manifestations of MACV, their lack of hemorrhagic manifestations and variable mortality rates (particularly with Carvallo) may limit their utility for testing vaccines and therapeutics. A summary of reported experimental MACV infections of hamsters of varying ages can be seen below in Table 10.

#### 3.2.4. Table 10: Experimental MACV Infection of Hamsters (*Cricetinae*)

**Table 10 microorganisms-13-01358-t010:** Table of studies of experimental MACV infection of hamsters (*Cricetinae)*, displaying age of macaques used, animal strain (if applicable), viral strain and passage history, viral inoculum size and route of infection, reported clinical manifestations of infection, average time to death, mortality rate, and source.

Age, Animal Strain (Noted If Applicable)	Virus Strain, Passage History, Inoculum, Route of Infection	Clinical Disease	Time to Death (Days Post-Infection)	Mortality Rate (%)	Source
Suckling hamster, aged less than 5 days	Carvallo (prototype), obtained directly from blood of infected individuals, unknown viral titer, i.p. or i.c.	Growth retardation; rough fur; dullness of fur coat; lack of coordination; tonic-clonic convulsions; ataxia; apnea; rigidity; underdevelopment	Unspecified	>0% (unspecified)	[92]
Juvenile hamster, aged greater than 4 weeks	Carvallo (prototype), obtained directly from blood of infected individuals, unknown viral titer, i.p. or i.c.	No clinical manifestations of infection observed	N/A	0% (unspecified)	
Suckling hamsters, aged less than 5 days	Carvallo (prototype), passaged twice in suckling hamster brain, 10^1.69^ PFU, i.c.	Not reported	7–11	100% (7/7)	[93]
	Cochabamba, passaged twice in suckling hamster brain, 10^1.02^ PFU, i.c.	Non-suppurative encephalitis	10–17	100% (6/6)	
Suckling hamsters, aged less than 6 days	Carvallo (prototype), passaged eight or fewer times in suckling hamster or suckling mice, unknown viral titer of inoculum, i.c., i.p., or combined i.p./i.c.	Growth retardation; rough fur; dullness of fur coat; lack of coordination; tonic-clonic convulsions; ataxia; apnea; rigidity	Unspecified	>0% (unspecified)	[88,92]
Adult hamsters, aged more than 5 weeks	Carvallo (prototype), passaged eight or fewer times in suckling hamster or suckling mice, unknown viral titer of inoculum, i.c., i.p., i.n., or oral	No clinical manifestations of infection reported	N/A	0% (unspecified)	
Adult hamsters, aged between 5 and 6 weeks	Carvallo (prototype), passaged twice in hamsters (presumed suckling hamster brain), 10^4^ Suckling Hamster LD_50_				[85]

#### 3.2.5. Guinea Pigs (*Cavia porcellus*)

Guinea pigs are one of the most thoroughly studied models for MACV infection, with research conducted on the prototype Carvallo strain, as well as the later-isolated Cochabamba and Chicava strains [86,94]. Broadly, MACV-Carvallo infection of guinea pigs is reported to result in highly variable, dose-independent mortality (20–80%), but uniform mortality following infection at doses as low as 2 PFU can reportedly be obtained following five passages in guinea pig spleens [23]. Transmission between guinea pigs in neighboring cages has been reported to occur, pointing to possible aerosol transmission, but the use of filtered cages and initial disinfection of bedding prior to cage cleaning was reported to eliminate this transmission [23].

Infection of C-13 strain guinea pigs is lethal, but few clinical signs are observed [95]. Uniform, dose-independent mortality was observed in guinea pigs given aerosolized MACV-Chicava [94]. Initial clinical signs observed included piloerection, fever, appetite loss, skin erythema, dyspnea, and intermittent, sometimes bloody diarrhea [94]. A later neurological phase (16–20 days post-infection) was associated with head tilt, ataxia, rapid breathing, respiratory difficulties, and weight loss [94]. Post-mortem analysis revealed multifocal petechial hemorrhages on the uterine and gastrointestinal surfaces, hepatic pallor, enlarged lymph nodes, and non-collapsible lungs [94]. Histopathology showed inflammation in multiple organs, including the liver, adrenal glands, and brain, with associated gliosis, meningitis, and perivascular inflammation [94]. Infection of Hartley guinea pigs at a higher dose of Chicava (10,000 PFU) i.p. led to similar outcomes (i.e., uniform mortality due to meeting euthanasia criteria), with additional clinical signs like vomiting and occasional hind-limb paralysis [96].

In outbred suckling guinea pigs (less than 5 days old), infection with Carvallo and Cochabamba strains resulted in minimal reported clinical signs, primarily weight loss and inactivity, with death by natural infection occurring within 18–23 days [86]. Carvallo caused significantly higher mortality than Cochabamba [86]. Specifically, i.c. infection with 1.4 × 10^4^ PFU of the Carvallo strain resulted in 87.5% mortality, compared to 17% for the Cochabamba strain [86]. This pattern was confirmed in adult guinea pigs infected i.p., as 67% of Carvallo-infected guinea pigs died compared to 0% of the Cochabamba-infected ones [86]. Further experiments showed a strain-independent, dose-dependent mortality response in suckling guinea pigs, though Cochabamba death rates were lower across the board [86].

Guinea pigs appear to have significant utility as an MACV animal model of infection. High MACV-associated mortality rates in Hartley guinea pigs following Carvallo or Chicava infection point to the possibility of using Hartley guinea pigs as a model for testing the efficacy of MCMs. Further research is needed to clarify how different MACV strains induce coagulopathies in this model—particularly Carvallo and Cochabamba—since these strains generally do not produce pronounced hemorrhagic or neurological clinical signs, yet can still cause high mortality (especially the Carvallo strain). A summary of reported experimental MACV infections of guinea pigs of varying ages and strains can be seen below in Table 11.

#### 3.2.6. Table 11: Experimental MACV Infection of Guinea Pigs (*Cavia porcellus*)

**Table 11 microorganisms-13-01358-t011:** Table of studies of experimental MACV infection of guinea pigs, displaying age of guinea pigs used, animal strain (if applicable), viral strain and passage history, viral inoculum size and route of infection, reported clinical manifestations of infection, average time to death, mortality rate, and source.

Age, Animal Strain (Noted If Applicable)	Virus Strain, Passage History, Inoculum, Route of Infection	Clinical Disease	Time to Death (Days Post-Infection)	Mortality Rate (%)	Source
Adult C-13 guinea pigs (inbred), age unspecified	Unspecified strain, unknown passage history and viral titer; undisclosed route of infection	Reportedly nondescript; no reported hemorrhagic or neurological manifestations	Unspecified	>0% (unspecified)	[95]
Adult Hartley (outbred) guinea pigs, age unspecified	Carvallo (prototype), second or third passage from suckling hamster brain, 1.4 × 10^4^ PFU, i.p.	Weight loss, inactivity	18–23	84% (21/25)	[86]
	Cochabamba, second passage from suckling hamster brain	Not reported	N/A	0% (0/6)	
Suckling or Adult Hartley (outbred) guinea pigs, aged 5 days or less or unspecified age, respectively	1000 PFU Carvallo (suckling), i.p.	Weight loss, inactivity	18–23	84% (21/25)	
	100 PFU Carvallo (suckling), i.p.			64.7% (11/17)	
	10 PFU Carvallo (suckling), i.p.			100% (11/11)	
	1 PFU Carvallo (suckling), i.p.			33% (6/18)	
	1.4 × 10^4^ PFU Carvallo (adults), i.p.			67% (4/6)	
	6.6 × 10^5^ PFU Cochabamba (adults), i.p.			0% (0/6)	
	Unspecified PFU, Carvallo (suckling), i.c.			87.5% (49/56)	
	Unspecified PFU, Cochabamba (suckling), i.c.			17.2% (5/29)	
	7000 PFU Cochabamba (suckling), i.p.			30.4% (7/23)	
	700 PFU Cochabamba (suckling), i.p.			16.6% (2/12)	
	70 PFU Cochabamba (suckling), i.p.			20% (3/15)	
	7 PFU Cochabamba (suckling), i.p.			4.5% (1/22)	
Dunkin-Hartley (outbred) guinea pigs of unspecified age	Chicava, passaged twice in Vero E6 cells, 10, 100 or 1000 PFU, aerosolized	Piloerection, fever, loss of appetite, erythema of haired skin (axillary, inguinal, ear tips), dyspnea, intermittent diarrhea (possibly bloody). Neurological signs—16–20 days post-infection, head tilt and ataxia; rapid breathing and respiratory difficulties; weight loss.	<30	100% (10/10)	[94]
Young adult Hartley (outbred) guinea pigs, 6–8 weeks of age	Chicava, passaged twice in Vero E6 or Neuro-2A cells, 10^4^ PFU, i.p.	Mild fever, weight loss, vomiting and hind-limb paralysis (in some cases)	19–22	100% (4/4)	[96]
Guinea pigs, unspecified age and strain	Carvallo, passaged five times in guinea pig spleen, 2 PFU (and higher, unspecified doses), unspecified route of infection	Not reported	Not reported	100% (unspecified)	[23]

#### 3.2.7. Geoffroy’s Tamarins (*Sanguinus geoffroyi*)

Geoffroy’s tamarins, along with guinea pigs, were among the first models of MACV infection tested. Infection with the Carvallo strain consistently results in the natural death of Geoffroy’s tamarins, making them a good model for developing vaccines or treatments [88]. However, the pathology of MACV infection in these animals is not well-characterized. After s.c. inoculation with 10^4^ hamster LD_50_ of MACV Carvallo, tamarins showed signs of inactivity, reduced appetite, general weakness, tremors, clinical shock, and a drop in body temperature 1–3 days before death [88]. Virus was recovered from the brain, spleen, kidneys, heart, and liver, with all but one animal dying within 12 days [88].

Higher doses (10^5.7^ hamster LD_50_) diluted at various factors resulted in death within 8–20 days, with higher dilutions resulting in a prolonged time-to-death [88]. Effective viral delivery methods included corneal instillation and application to scarified skin; infection through i.n. and oral routes did not result in observable clinical signs [88]. Viral loads were detected in the blood and various organs, with lymph node cortical necrosis and splenic reticular hyperplasia also noted [95].

While Geoffroy’s tamarins show promise as a model for studying MACV due to their high mortality rate, further research is needed to better characterize MACV pathogenesis in these animals before their utility in this field is fully assessed. A summary of reported experimental MACV infections of Geoffroy’s tamarins can be seen below in Table 12.

#### 3.2.8. Table 12: Experimental MACV Infection of Geoffroy’s Tamarin (*Sanguinus geoffroyi*)

**Table 12 microorganisms-13-01358-t012:** Table of studies of experimental MACV infection of Geoffroy’s tamarins, displaying age of tamarins used, animal strain (if applicable), viral strain and passage history, viral inoculum size and route of infection, reported clinical manifestations of infection, average time to death, mortality rate, and source.

Age, Animal Strain (Noted If Applicable)	Virus Strain, Passage History, Inoculum, Route of Infection	Clinical Disease	Time to Death (Days Post-Infection)	Mortality Rate (%)	Source
Undisclosed age	Carvallo (prototype), passaged a maximum of eight times in suckling hamsters or suckling mice, 10^5.7^ suckling hamster LD_50_, or 1:10, 1:100, 1:1000, 1:10,000. 1:100,000, or 1:1,000,000 dilutions thereof, s.c.	Sick appearance, anorexia, weakness, inactivity, decrease in bodily temperature 1–3 days prior to death	11 (undiluted dose)	100% (2/2)	[88]
			13.5 (1:10 dilution)		
			13 (1:100 dilution)		
			18 (1:1000 dilution)		
			20 (1:10,000 dilution)		
			16 (1:100,000 dilution)		
		Undisclosed (1:1,000,000 dilution)	N/A	0% (0/2)	
	Carvallo (prototype), passaged a maximum of eight times in suckling hamsters or suckling mice, 10^5.7^ suckling hamster LD_50_, application to scarified skin or corneal instillation	Sick appearance, anorexia, weakness, inactivity, decrease in bodily temperature 1–3 days prior to death	17 (application to scarified skin)	100% (3/3)	
			Unspecified (corneal instillation)	33% (1/3)	
	Carvallo (prototype), passaged a maximum of eight times in suckling hamsters or suckling mice, 10^5.7^ suckling hamster LD_50_, i.n. or oral	No reported clinical signs	N/A	0% (unspecified)	
Unspecified age	Unspecified strain and passage history	Anorexia, tremors, shock	8–20	>0% (unspecified)	[88,95]

#### 3.2.9. African Green Monkeys (*Chlorocebus aethiops*) (AGMs)

There are two public reports of the experimental MACV infection of African green monkeys—one focusing on clinical course and the other on pathology following infection of the same monkeys [97,98]. Six young adult AGMs were inoculated s.c. with 1000 PFU of MACV-Carvallo [97]. Five AGMs naturally died between 10 and 13 days post-infection, with the first observed clinical signs appearing 3–4 days post-infection [97,98]. The disease progressed beyond fever to include depression, anorexia, dehydration, conjunctivitis, nasal discharge, and bleeding from the gums, nares, and rectum [97]. The sole AGM survived the initial hemorrhagic phase of disease but developed neurological signs on day 18, specifically lack of coordination and tremors, and ultimately led to emaciation and natural death at day 24 [97].

A later report summarizing gross and microscopic pathological lesions associated with MACV-Carvallo infection in AGMs revealed a variety of lesions at different stages of infection [98]. Moderate-to-severe, acute suppurative broncho-pneumonia was observed across all AGMs, marked by pulmonary lesions with hemorrhaging and inflammation [98]. In three AGMs, co-localization of Gram-negative rod-shaped bacteria with these pulmonary lesions was noted, suggesting that secondary bacterial infections are common [98]. In some or all AGMs that died during the acute, hemorrhagic phase of infection, the following gross pathological lesions were recorded: hemorrhage in the lungs, thymus, lymph nodes, subcutis, oral cavity, and intestines; enlargement of the spleens and lymph nodes; and enteritis [98].

Microscopic pathological lesions in some or all AGMs that succumbed included moderate-to-severe hepatic necrosis; minimal-to-severe necrosis of the intestines; minimal-to-moderate skin and oral mucosal necrosis; mild-to-severe necrosis of the adrenal glands; and minimal-to-mild lymphoid tissue necrosis in the tonsils and the thymus [98]. Three AGMs displayed mild-to-severe pancreatitis, associated with pancreatic lesions including mononuclear cell infiltration, dilation of acinar glands, and small ductule formation [98]. Additionally, microscopic examination revealed minimal-to-moderate degrees of hemorrhaging in four of five fatal AGMs, largely in the liver, the lamina propria and submucosa of the small and large intestine, and in the adrenal glands [98]. One AGM also displayed signs of an ileo-colic intussusception [98]. Regardless of time of death, all AGMs displayed swelling of liver cells and fatty changes [98]. Mild-to-severe acute thrombosis, including one case in the brain, was also observed in four of five monkeys [98]. The AGM with brain lesions also exhibited signs of severe encephalomyelitis [98]. Reported histological lesions in this animal included gliosis, lymphocyte infiltration, and lymphoreticular cuffing in the brain stem, which were also present in both white and gray matter [98].

These results indicate that AGMs can effectively recapitulate both the hemorrhagic and neurological manifestations of MACV infection seen in humans and are thus a valuable model for studying BHF pathogenesis and testing MCMs [97].

#### 3.2.10. Cynomolgus Macaques (*Macaca fascicularis*)

Cynomolgus macaques are also an effective animal model for BHF. S.C. infection with MACV-Carvallo typically follows a biphasic course, like rhesus macaques, with an acute phase followed by a neurological phase [99]. MACV-Chicava infection (i.m. or aerosol) leads to the presentation of neurological clinical signs concurrently with the hemorrhagic acute phase [100]. Regardless of strain or route of infection, MACV infection in cynomolgus macaques leads to high natural mortality rates [99,100].

MACV-Carvallo infection (1000 PFU, s.c.) results in 86% mortality and clinical signs similar to those seen in rhesus macaques, including conjunctivitis, depression, anorexia, fever, and sporadic diarrhea [99]. However, clinical signs in cynomolgus macaques are generally less severe, especially before death, as worsening anorexia and depression are less pronounced [99]. About 29% of the animals survive the initial phase, but half of these survivors succumb during the neurological phase, characterized by tremors, nystagmus, lack of coordination, paresis, and coma [99].

When challenged with a s.c. dose of 3000 focus-forming units (FFU) MACV-Carvallo, all infected macaques met euthanasia criteria by day 20, showing clinical signs of febrile illness followed by hemorrhagic manifestations, including epistaxis, petechiae, and gastrointestinal bleeding [101,102]. Severe hemorrhage in the lungs, gastrointestinal tract, and intracranial space, along with acute hepatitis, were common [101]. Coagulation abnormalities were observed, with prolonged clotting times and decreased levels of clotting factors [101].

These studies confirm that cynomolgus macaques are a valuable model for studying MACV pathogenesis and developing MCMs despite slight differences from human disease (i.e., biphasic progression) [99,100]. Indeed, cynomolgus macaques are often used to evaluate the efficacy of candidate vaccines and therapeutics [102]. A summary of reported experimental MACV infections of cynomolgus macaques of varying ages can be seen below in Table 13.

#### 3.2.11. Table 13: Experimental MACV Infection of Cynomolgus Macaques (*Macaca fascicularis*)

**Table 13 microorganisms-13-01358-t013:** Table of studies of experimental MACV infection of cynomolgus macaques, displaying age of macaques used, animal strain (if applicable), viral strain and passage history, viral inoculum size and route of infection, reported clinical manifestations of infection, average time to death, mortality rate, and source of study utilized.

Age, Animal Strain (Noted If Applicable)	Virus Strain, Passage History, Inoculum, Route of Infection	Clinical Disease	Time to Death (Days Post-Infection)	Mortality Rate (%)	Source
Adult macaques, age unspecified	Carvallo (prototype), third/fourth passage in suckling hamster brain, 1000 PFU, s.c.	Initial phase: conjunctivitis, depression, anorexia, fever, dehydration, initial constipation followed by diarrhea, nasal discharge (at times bloody), clonic spasms Neurological phase: Severe intention tremors, nystagmus, lack of coordination, paresis, coma	17.0	6/7 (86%) (total, both phases of infection)	[99]
				5/7 (71%) (initial phase of infection)	
				1/2 (50%) (neurological phase of infection)	
Adult male macaques, aged 2.5 years	Carvallo (prototype), passaged in Vero E6 cells, 3000 FFU, s.c.	Epistaxis, petechiae, non-specific febrile clinical signs, melena, hematochezia, balance disorders, reduced activity, dehydration, weight loss	11–18	6/6 (100%)	[101,102]
Adult macaques, age unspecified	Chicava, passaged twice in Vero E6 cells, 1000 PFU, i.m.	Reduced appetite, weakness, depression, mild fever, dehydration, rash (axillary and inguinal regions), hematuria, epistaxis. Tremors, ataxia, facial edema, darkened hard plate, black/purple discoloration of the lips	18.3	100% (4/4)	[100]
	Chicava, passaged twice in Vero E6 cells, 100 PFU, aerosolized	Similar clinical signs as in i.m. group, inclusion of mild-moderate nonsuppurative encephalitis/gliosis	17.3	75% (3/4)	
	Chicava, passaged twice in Vero E6 cells, 1000 PFU, aerosolized		19.5	50% (2/4)	
	Chicava, passaged twice in Vero E6 cells, 10,000 PFU, aerosolized		21.3	75% (3/4)	

#### 3.2.12. Rhesus Macaques (*Macaca mulatta*)

Rhesus macaques are a well-characterized model for MACV infection. The clinical course of MACV infection in rhesus macaques proceeds as an initial acute phase resolving around 20 days post-infection, followed by a neurological phase beginning 26–30 days post-infection and concluding near day 40 [99,103]. The acute phase is marked by fever, depression, anorexia, dehydration, diarrhea, conjunctivitis, clonic spasms, nasal discharge, rash, petechiae, bleeding gums, hypothermia, and hypotension [99,103,104,105]. Roughly 80% of rhesus macaques infected with MACV-Carvallo died naturally during the acute phase of infection, while the remainder of macaques (predominantly more mature macaques) had an initial resolution but eventually died naturally during the late neurological phase of infection [23,99]. Infected individuals in the neurological phase present with a lack of coordination, severe tremors, mucopurulent nasal discharge, alopecia, diarrhea, emaciation, paresis, convulsions, paralysis, muscle atrophy, dermatitis, and nystagmus [99,103].

Pathologically, MACV infection in rhesus macaques is associated with hepatic necrosis, gastrointestinal epithelial necrosis, adrenal cortical necrosis, and lymphoid depletion [105]. Less common findings include interstitial pneumonia, pulmonary edema, and central nervous system vasculitis [105]. Key hematological findings include thrombocytopenia; the presence of fibrin split products; elevations in plasma fibrinogen and sorbitol dehydrogenase levels; reductions in serum albumin levels; and elevated activated partial thromboplastin time [106]. The outcome of infection is dose-dependent, with lower dose infections (10 PFU) resulting in a 25% mortality rate (natural death by infection), and higher dose infections (10^3^ and 10^5^ PFU) leading to near-total mortality (97.5–100% natural death by infection) [99,103,104,105]. Most macaques die during the acute phase, with a smaller proportion succumbing during the neurological phase [99,103]. One study reported a total mortality rate of 96% at a dose of 10^3^ PFU [99].

Although there are differences between the disease progression in macaques and humans—such as more frequent hepatic necrosis in macaques and the absence of a distinct neurological phase in humans—rhesus macaques remain a valuable model for studying MACV [105]. Their high mortality rates and the similarities in pathologies with human infections make them useful for researching pathogenesis and developing MCMs. Using this model, researchers have investigated antibody treatments, ribavirin, a live attenuated MACV Carvallo vaccine (created through serial passaging), and cross-protection conferred by the Candid#1 vaccine [23,107,108]. A summary of reported experimental MACV infections of rhesus macaques of varying ages can be seen below in Table 14.

#### 3.2.13. Table 14: Experimental MACV Infection of Rhesus macaques (*Macaca mulatta*)

**Table 14 microorganisms-13-01358-t014:** Table of studies of experimental MACV infection of rhesus macaques, displaying age of macaques used, animal strain (if applicable), viral strain and passage history, viral inoculum size and route of infection, reported clinical manifestations of infection, average time to death, mortality rate, and source.

Age, Animal Strain (Noted If Applicable)	Virus Strain, Passage History, Inoculum, Route of Infection	Clinical Disease	Time to Death (Days Post-Infection)	Mortality Rate (%)	Source
Young adult macaques, unspecified age	Carvallo (prototype), third/fourth passage in suckling hamster, 1000 PFU s.c.	Initial phase: conjunctivitis, depression, anorexia, fever, dehydration, initial constipation followed by diarrhea (majority), nasal discharge (at times bloody) (half), clonic spasms, erythematous facial and abdominal rash, terminal progressive anorexia, dehydration and depression Neurological phase: severe intention tremors, nystagmus, lack of coordination, paresis, coma	19.3	45/47 (96%) (total macaques, initial and neurological phases of illness)	[99]
				39/47 (83%) (total macaques, initial phase)	
				6/8 (75%) (total macaques, neurological phase of illness)	
Mature macaques, unspecified age			30.5	2/4 (50%) (mature, 5–8 kg rhesus macaques, initial phase of illness)	
				37/43 (86%) (young, 2.5–4 kg rhesus macaques, initial phase of illness)	
				2/2 (100%) (mature, 5–8 kg rhesus macaques, neurological phase of illness)	
				6/8 (75%) (young, 2.5–4 kg rhesus macaques, acute phase of illness)	
Young adult macaques, age unspecified	Carvallo (prototype), third/fourth passage in suckling hamster, 10^3^ PFU s.c.	Skin petechiae, exudative rash, epistaxis	19.5	100% (4/4)	[105]
	Carvallo (prototype), third/fourth passage in suckling hamster, 10 PFU s.c.		37	25% (1/4)	
	Carvallo (prototype), third/fourth passage in suckling hamster, 10^5^ PFU s.c.	Not recorded	14.5	100% (4/4)	
	Carvallo (prototype), passaged in suckling hamster brain, 1000 PFU s.c.	Initial: diarrhea, depression, anorexia, necrosis of the skin Neurological: lack of coordination, mucopurulent nasal discharge, alopecia, diarrhea, emaciation, tremors, paresis and convulsions, lateral curvature of the spine, paralysis, muscle atrophy, dermatitis of the skin and abdomen, weakness	36.8	100% (4/4)	[103]
	Carvallo (prototype), third passage in suckling hamster brain, 10^3^ PFU s.c.	Fever, depression, anorexia, diarrhea, clonic spasms, adipsia, nasal discharge, facial and abdominal rashes, dehydration, purulent nasal discharge, bleeding from the nares and skin, terminal hypothermia	19.5	100% (4/4)	[104]
	Carvallo (prototype), third passage in suckling hamster, 10^5^ PFU s.c.		14.3		
	Carvallo (prototype), third passage in suckling hamster brain, 10 PFU s.c.	No reported clinical signs	N/A	0% (0/4)	
	Carvallo (prototype), fourth passage in suckling hamster brain, 1000 PFU s.c.	Lethargy, decreased appetite, dehydration, epistaxis, skin petechiae, bleeding gums, terminal hypotension and hypothermia	13–21	100% (14/14)	[106]

### 3.3. Summary of MACV Animal Models

Several animal models have been used to study BHF, each offering unique advantages and drawbacks. Rhesus and cynomolgus macaques generally provide the most faithful replication of severe disease, showing both hemorrhagic and neurologic phases alongside high mortality, but they require specialized care, have high costs, and raise ethical concerns. AGMs also display robust disease with both hemorrhagic and neurological features, though reports are limited to a small number of published studies. Geoffroy’s tamarins show high lethality and can be infected by multiple routes, yet their pathology and disease course are less completely characterized. Among rodent models, guinea pigs are the most extensively characterized and can exhibit lethal infections (particularly certain strains like Carvallo or Chicava), making them a popular platform for evaluating MCMs; however, their disease course is sometimes strain- and dose-dependent, and hemorrhagic signs may not be as pronounced as in primates. Hamsters, especially when very young, are susceptible to certain MACV strains and can develop neurological signs, but infections in older hamsters are often mild or asymptomatic, limiting their utility. Mice—particularly suckling or immunodeficient strains—can experience high mortality from MACV, which is advantageous for mechanistic and therapeutic studies, but they rarely show the full hemorrhagic clinical picture and thus can only partially mimic human BHF.

## 4. Guanarito Virus (GTOV)

### 4.1. Background

Guanarito virus was first reported following a September 1989 outbreak of hemorrhagic illness of then-unknown etiology emerging in the southeastern part of the Venezuelan state of Portuguesa [109]. Initially thought to be dengue fever, later analysis indicated the causative agent of Venezuelan Hemorrhagic Fever (VHF) was a previously uncharacterized arenavirus [109]. Subsequently, 1–8 monthly cases were reported in the same region and in rural areas of neighboring Barinas state until 1992 [110]. Clinical symptoms of 15 patients infected with GTOV in Portuguesa state in 1990–1991 were tracked over the course of infection [109]. Most or all exhibited fever, prostration, arthralgia, headache, dehydration, pharyngitis, diarrhea, somnolence and conjunctivitis; in multiple cases, cough, nausea, vomiting, epistaxis, bleeding gums, menorrhagia, facial edema, tonsillar exudate, sporadic pulmonary crackles, and cervical lymphadenopathy were noted; in single cases, hematemesis, abdominal pain, chest pain, vertigo, convulsions, hepatomegaly, hand tremors, and rash were prominent [109]. Later research revealed that the most common symptoms of GTOV infection are malaise, fever, headache, bleeding of gums, and arthralgia [110]. VHF incidence subsequently declined from September 1992–August 1996, with 9 total cases occurring during this period, but another surge occurred between August 1996 to May 1997, with 1–16 cases reported monthly, resulting in 165 total VHF cases between 1989 and 1997 [110]. Later epidemiological surveillance work/case reports indicated that 618 VHF cases had occurred over 1989–2006 in Portuguesa state, with a 23.1% case fatality rate [111]. Since 2006, surveillance and epidemiology work in Venezuela has been limited, making it nearly impossible to compile accurate case counts for the ensuing period, particularly from 2006–2021 and 2022–2025 [4]. A recent publication indicated that VHF cases have almost entirely occurred in the llanos of Venezuela, including in areas beyond previously reported zones of known viral endemicity within Portuguesa and Barinas states, such as the Calabozo municipality of Guárico state [112]. Other regions include multiple municipalities in Apure, Portuguesa, and Barinas states, as well as in an Andean state, Trujillo [112]. More recent reports have concluded a total of 118 suspected cases of VHF for most of 2021 (up to the time of publication), with 36 confirmed as GTOV [113]. A total of 20 VHF cases were reported in Portuguesa state (8/20 confirmed to be GTOV), 94 cases were reported in Barinas state (28/94 confirmed to be GTOV), and 4 cases were reported in Apure state (0/4 confirmed to be GTOV) [113]. No further publications or case reports have since been released.

A field epidemiological study identified *Sigmodon alstoni* and *Zygodontomys brevicauda* as GTOV reservoirs and posited that human exposure likely stems from exposure to infected rodent excreta [114]. This study indicated that the two species respond to GTOV infection differently, as the majority of *Zygodontomys brevicauda* produced GTOV-specific antibodies, unlike *Sigmodon alstoni* [114]. Experimental infection of newborn, juvenile, and adult cane mice showed persistent, asymptomatic infection characterized by chronic viremia and virus shedding in urine/secretions in most adults (the latter which could take 18 days to clear, the former of which cleared between 17-63 days post-infection) [115].

### 4.2. Animal Models of Experimental Infection with Guanarito Virus

To date, CD-1 strain outbred mice, rhesus macaques, cynomolgus macaques, and guinea pigs (both Strain 13 and Hartley) have been reported in the literature as experimental models of VHF. These species have differential responses to infection with the prototype strain, as outlined below in Figure 7.

#### 4.2.1. Mice (*Mus musculus*)

In a 1994 study, CD-1 strain (outbred) suckling mice (i.e., 3–6 days old) were inoculated i.c. with 25 µL of GTOV stock of unknown titer prepared from infected Vero cells [116]. Clinical sign onset was noted within 10 days post inoculation, at which time lethargy, ataxia, runting, and hind-limb paralysis were observed [116]. A majority of the mice naturally died, with deaths observed after the twelfth day post-infection and sporadically thereafter [116]. By contrast, when adult mice of the same strain were inoculated i.p. with a 10% stock suspension of infected baby mouse brain homogenate, no clinical signs of infection were observed [116]. Given the lethality observed, CD-1 suckling mice may be a worthwhile model for recapitulating the neurological manifestations of GTOV, whereas adult mice are not well-suited.

#### 4.2.2. Guinea Pigs (*Cavia porcellus*)

Guinea pigs are the best-characterized GTOV animal model of infection, with experiments involving both Strain 13 and Hartley guinea pigs. The first study characterizing GTOV in guinea pigs was performed in Strain 13 guinea pigs following s.c. infection with 10^3.4^ PFU GTOV [116]. All guinea pigs naturally died between 11 and 14 days post-infection [116]. In another study, Strain 13 and Hartley guinea pigs were inoculated s.c. with a GTOV titer ranging between 1.0 and 3.4 log_10_ PFU [117]. Following euthanasia due to moribundity, histopathological examination of Strain 13 guinea pigs revealed interstitial pneumonia, diffuse fatty changes in the liver, splenic congestion and lymphoid necrosis, adrenal congestion, and zona fasciculata necrosis [117]. Half of this cohort developed epithelial necrosis in the small intestine and colon; pancreatic acini-cytoplasmic vacuolization; renal tubule regeneration; lymphoid necrosis in the lymph nodes; hemorrhage formation in the lung; and focal necrosis in the liver [117]. In most Hartley guinea pigs, diffuse fatty changes in the liver, interstitial pneumonia, lymphoid necrosis in the lymph nodes, testicular atrophy, and bone marrow necrosis were all observed following euthanasia due to moribundity [117]. Less common histological lesions included chronic choroid plexus inflammation, chronic cardiac multifocal inflammation, tracheal focal epithelial necrosis, lung hemorrhage formation, multifocal or centrilobular necrosis in the liver, chronic hepatic inflammation, lymphoid depletion in the lymph nodes, cortical necrosis in the thymus, bone marrow depletion, adrenal gland hemorrhage and congestion, epithelial necrosis and hemorrhage in the stomach, epithelial necrosis in the colon, small intestine, colon, and cecum, and esophageal basal cell increased mitoses and epithelial necrosis [117]. An i.p. challenge of 16 Hartley guinea pigs with 2000 PFU of GTOV confirmed the presence of these same lesions [118]. During a 12-day period of monitoring following infection, guinea pigs lost on average 7.5% of their body weight and displayed relatively mild gross pathological findings—namely, areas of red discoloration in the lungs, multifocal areas of tan liver discoloration, and enlarged mandibular and mesenteric lymph nodes [118].

As guinea pigs do not fully replicate GTOV’s hemorrhagic manifestations, further research is needed to clarify the virus’s pathogenesis in this model—especially regarding coagulopathy. Still, their high mortality rates make them valuable for evaluating MCM efficacy, as evidenced by preclinical trials of a GTOV-targeted antibody treatment [119]. A summary of reported experimental GTOV infections of various rodent species can be seen below in Table 15.

#### 4.2.3. Table 15: Successful Rodent Animal Models of Experimental GTOV Infection

**Table 15 microorganisms-13-01358-t015:** Table of successful types of rodent animal models of experimental GTOV infection, listed by species, animal strain/age, viral strain, passage history and inoculum size/route, clinical signs, average time to death, mortality rate, and source.

Species, Age, Animal Strain (Noted If Applicable)	Virus Strain, Passage History, Inoculum, Route of Infection	Clinical Disease	Time-to-Death (Days Post-Infection)	Mortality Rate (%)	Source
Suckling CD-1 (outbred) mice (*Mus musculus*), aged 3–6 days	INH-95551 (prototype), passaged once in Vero cells, unknown viral titer, i.c.	Lethargy, ataxia, runting, hind-limb paralysis	12+	>0% (majority, unspecified)	[116]
Adult mice (*Mus musculus*), age/strain unspecified	INH-95551 (prototype), 10% suspension of infected suckling mouse brain, unknown viral titer, i.p.	None reported	N/A	0% (0/unspecified)	
Adult strain 13 (inbred) guinea pig (*Cavia porcellus*), age unspecified	INH-95551 (prototype), passaged twice in suckling mouse brain, 10^3.4^ PFU, s.c.	Minimal hemorrhagic manifestations, moribund state	11–14	100% (10/10)	
	INH-95551 (prototype), passaged twice in suckling mice brains and once in Vero cells, 1.0–3.4 log10 PFU, s.c.		12–14 (killed when moribund)	100% (2/2)	[117]
Adult Hartley (outbred) guinea pig (*Cavia porcellus*), age unspecified				100% (6/6)	
Hartley (outbred) guinea pig (*Cavia porcellus*)	INH-95551 (prototype), passaged in Vero cells, 2000 PFU, i.p.	Weight loss	N/A	0% (0/16)* *(animals sacrificed for pathology data collection)	[118]
		Weight loss, elevated body temperature	15	100% (unspecified)	[119]

#### 4.2.4. Cynomolgus Macaques (*Macaca fascicularis*)

The experimental infection of cynomolgus macaques with GTOV was reported in two separate publications in 2023. Both reports involved the same set of cynomolgus macaques [101,102]. Six adult, female cynomolgus macaques were inoculated s.c. with 3000 FFU of the prototype strain [102]. Three macaques received a Mopeia virus-vectored vaccine, whereas three controls remained unvaccinated [102]. The first clinical signs in unvaccinated macaques were evident as soon as 2 days post-infection, including reduced activity, weight loss, fever, and gastrointestinal issues [102]. Untreated macaques also developed epistaxis, petechiae, melena, and hematochezia [101]. Treated animals failed to develop any notable clinical manifestations of infection [102]. One untreated animal met endpoint criteria at 14 days post-infection, but the two other untreated animals survived through the 39-day study period [102]. All animals showed signs of mild acute hepatitis, characterized by swollen hepatocytes and relatively diffuse councilman apoptotic bodies [101]. The macaque that died at 14 days post-infection also displayed signs of acute lung injury, both through parietal thickening and hemorrhaging in the lung parenchyma [101]. Hematological analysis revealed elevated activated partial thromboplastin time and prothrombin time in infected animals by day 8 post-infection, but returning to baseline by day 12 post-infection [101]. Fibrinogen levels rose by day 8 post-infection and stayed elevated through day 12 [101]. Likewise, the activity of coagulation factors XI, IX, VIII, and VII declined, although these decreases were transient for all but factor VIII. Most factors reached their nadir near day 8 but recovered by day 12—except for VIII, which remained low [101]. Based on these pathological findings, cynomolgus macaques are decently suited for pathogenesis and immunology studies, but less ideal for the development of MCMs due to incomplete lethality in this model. Still, more work is needed to better characterize the potential of cynomolgus macaques as an animal model of GTOV infection.

#### 4.2.5. Rhesus Macaques (*Macaca mulatta*)

A 1994 study was the first and only report of experimental GTOV infection of rhesus macaques [116]. Three adult rhesus macaques received a 10^3.4^ PFU s.c. inoculation of the prototype strain of GTOV [116]. These macaques developed various clinical manifestations, including lethargy, reduced appetite, and fever. Transient viremia was reported between days 4–18 post-infection [116]. All animals ultimately recovered completely from infection and produced high levels of specific neutralizing antibodies [116]. Because rhesus macaques experience only mild illness when infected with GTOV, they are not generally recommended as a primary model for studying severe pathogenesis or for vaccine/antiviral development. However, they do generate a robust immune response, which may be valuable for other research applications. A summary of reported experimental GTOV infections of various NHP species can be seen below in Table 16.

#### 4.2.6. Table 16: Successful NHP Animal Models of Experimental GTOV Infection

**Table 16 microorganisms-13-01358-t016:** Table of successful types of NHP animal models of experimental GTOV infection, listed by common name and species, age of model used, viral strain and passage history, animal strain, viral inoculum size and route of infection, reported clinical manifestations of infection, average time to death, mortality rate, and source.

Species, Age, Animal Strain (Noted If Applicable)	Virus Strain, Passage History, Inoculum, Route of Infection	Clinical Disease	Time-to-Death (Days Post-Infection)	Mortality Rate (%)	Source
Adult rhesus macaque (*Macaca mulatta*), age unspecified	INH-95551 (prototype), passaged twice in suckling mice brains, log_10_ 3.4 PFU, s.c.	Lethargy, reduced appetite, fever	N/A	0% (0/3)	[116]
Adult cynomolgus macaque (*Macaca fascicularis*), aged 3 years	INH-95551 (prototype), passaged once in Vero cells, 3000 FFU, s.c.	Weight loss, reduced activity, fever, epistaxis, petechiae, melena, hematochezia	11–18	33.3% (1/3)	[101,102]

### 4.3. Summary of GTOV Animal Models

Studies involving GTOV—the causative agent of VHF—have relied on NHPs and rodent models to investigate disease mechanisms and test potential MCMs. Rhesus macaques experience only mild clinical illness and survive infection, making them less ideal for modeling severe GTOV disease, though their immune response could be useful for other research studies. Cynomolgus macaques more accurately mirror important infection features, including coagulopathy and moderate-to-severe clinical signs, but only one-third of infected animals may succumb, limiting their value for evaluating vaccines and treatments that require uniform lethality. Among rodents, guinea pigs (Strain 13 and Hartley) typically exhibit high mortality, showing hepatic and pulmonary involvement similar to human disease, but often with minimal hemorrhagic manifestations. Meanwhile, CD-1 suckling mice can develop severe neurological signs and experience significant mortality; however, adult mice tend to remain asymptomatic, reducing their applicability. Consequently, while cynomolgus macaques and guinea pigs capture important aspects of GTOV pathology and mortality, neither perfectly reproduces the hallmark hemorrhagic presentation, underscoring the need for further model refinements and comparative studies.

## 5. Chapare Virus (CHAPV)

### 5.1. Background

The first outbreak of Chapare virus was reported in late 2003 [2]. A cluster of hemorrhagic fever cases of then-unknown etiology was reported in a rural area near the Chapare River in the Cochabamba department of Bolivia [2]. The exact number of cases that occurred in this outbreak remains unclear, and the clinical symptoms associated with the course of infection were not recorded in all individuals [2]. A specific individual whose clinical course of infection was tracked initially presented with fever, headache, joint stiffness, muscle pain, and vomiting [2]. In the later stages of infection, hemorrhagic signs emerged, resulting in significant deterioration and patient death by 14 days post-symptom onset [2]. Viral samples from this case were preserved and analyzed through RT-PCR and phylogenetic analytical methods, revealing that the agent of infection was a previously unknown arenavirus, closely related to Sabiá virus [2]. This virus, subsequently named Chapare virus, was most recently associated with two clusters of cases over 2019–2020 [2]. A 2019 outbreak began in Caranavi, a municipality in the La Paz department of Bolivia, and subsequently spread to the city of La Paz [120,121]. There were five cases; three were lethal [121]. Fever, myalgia, arthralgia and headaches were observed in most patients within this cluster, followed by hemorrhagic signs (gingival, vaginal) and death, aligning with the initial symptoms in the first fully recorded case of CHAPV [120,121]. Neurological signs were observed in all but one patient, though the manifestations varied significantly, ranging from seizure to paraparesis [120]. Human–human transmission was observed in three of these five cases through suspected nosocomial infection [120,121]. Four more cases were detected in 2019 and 2020 outside of this cluster [120]. All cases proceeded with similar symptoms as previous outbreaks of infection, with one death reported [120]. None of these cases were transmitted nosocomially, and only two of the cases related to each other—a mother and her child in Caranavi (the other two were of agricultural workers in Alto Beni and Palos Blancos, respectively) [120]. Given detection of CHAPV RNA in *Oligoryzomys microtis*, it is possible that this rodent is a reservoir of CHAPV, and that, like other rodent reservoirs of NWAVs, it plays a critical role in the transmission of CHAPV to humans through human exposure to infected rodents/infected rodent excreta [120]. Viral persistence in humans may also play a role in CHAPV transmission, as evidenced by the RT-qPCR detection of viral RNA in the semen and whole blood of survivors of CHAPV infection from the 2019 outbreak for as long as 170 days post-symptom onset [120]. The isolation of CHAPV from another survivor from the same outbreak at 86 days post-symptom onset further points to long-term CHAPV viral persistence [120]. This phenomenon and its possible ramifications may warrant further investigation in extant CHAPV animal models.

### 5.2. Animal Models of Experimental Infection with Chapare Virus

To date, two animal models of infection of CHAPV infection have been reported—cynomolgus macaques and Strain 13 guinea pigs. The presentation in these two animal models differed following challenge with the prototype strain of CHAPV, 810419. Reported clinical manifestations of infection and associated pathologies are discussed below

#### 5.2.1. Guinea Pigs (*Cavia porcellus*)

CHAPV infection of guinea pigs has been reported in a single publication [122]. In this study, six adult, mixed gender (four female and two male) Strain 13 guinea pigs received a 4500 PFU i.p. challenge of CHAPV 810419. Uniform mortality (due to meeting euthanasia criteria) was reported. The two male guinea pigs experienced a more rapid disease progression, euthanized at day 7 post-infection due to weight loss and morbidity. By comparison, weight loss in female guinea pigs occurred over days 7 to days 14–16 post-infection), and most saw elevations in bodily temperature. A female guinea pig displayed terminal hind-limb paralysis.

Hematological analysis revealed lymphocytopenia, monocytopenia, and thrombocytopenia in all guinea pigs from day 7 post-infection until death. Fibrin deposition was also noted. Blood chemistry analysis identified elevated alanine and aspartate aminotransferase in serum, indicative of hepatic dysfunction, which lasted until euthanasia. Lastly, decreases in total protein and albumin levels in blood, indicative of vascular failure, were observed in all animals at day 7 post-infection until euthanasia.

Gross pathological examination revealed lesions indicative of hepatic dysfunction, gastrointestinal inflammation, and/or gastrointestinal hemorrhage. Histopathological analysis revealed that all female guinea pigs had varying levels of hepatic necrosis and degeneration, along with interstitial pneumonia, alveolar septum expansion, sinus histiocytosis in the axillary and/or inguinal lymph nodes, and mononuclear cell infiltrates in various organs, such as the liver and the spleen. Ulcerative enteritis or typhlitis, minimal pyelitis and cystitis in the kidney and bladder, and mild alterations of splenic composition and morphology (in both the red and white pulp) were additional findings in some guinea pigs. Gliosis was noted in one female [122].

Strain 13 guinea pigs may not uniformly exhibit neurological clinical signs that have been observed at times in human CHF cases. However, their uniform mortality suggests their utility in evaluating the efficacy of vaccine candidates and antivirals. Sex-specific differences in mean time-to-death may indicate that guinea pigs experience a sexually dimorphic response to CHAPV infection, but further investigation is required to be probative.

#### 5.2.2. Cynomolgus Macaques (*Macaca fascicularis*)

A recent study reported an i.v. challenge of four cynomolgus macaques with 10,000 PFU of CHAPV 810419 [123]. Three of four macaques survived until the study’s endpoint at day 35; one met humane endpoint criteria on day 12. All macaques experienced progressive weight loss over the first two weeks, with one macaque showing hypothermia by day 14. Other clinical signs included reduced appetite (beginning around days 8–10), diarrhea (days 11–13), emesis (one macaque at day 14), epistaxis (in two macaques, occurring around days 11–15), intention tremors (in one macaque from day 15 onward), ataxia (in two macaques at days 4 and 12), and dehydration (in one macaque at day 11).

Hematological analysis revealed leukocytopenia and lymphocytopenia in all animals between days 7–10, with recovery by day 21 in the surviving macaques. Erythrocytopenia persisted throughout the study, peaking at days 7 and 10, while thrombocytopenia was noted from days 4–10, with recovery by day 21. Elevated alanine and aspartate aminotransferase levels indicated hepatic dysfunction. C-reactive protein levels increased, indicating inflammation, while total protein, urea nitrogen, calcium, and albumin levels decreased at various points. Additionally, viremia was detectable between days 4–10 post-infection, and viral RNA persistence was noted in various tissues (e.g., inguinal/axillary lymph nodes, spleen, adrenal gland) in macaques that survived to day 35 and the macaque that met endpoint criteria at day 12 post-infection.

The macaque that died before day 35 showed below-average fat stores, absence of normal stomach and small intestine contents, multifocal mucosal ulcers in the colon, and lymphadenomegaly. Histopathology revealed hepatic inflammation, lymphocytic infiltration, necrosis, splenic lymphatic depletion, interstitial pneumonia, adrenal cortex necrosis, brain stem perivascular cuffing, gliosis, and gastrointestinal tract inflammation. The surviving macaques showed no significant gross lesions, with minimal splenic lesions and only one showing minor neurological lesions [123].

While cynomolgus macaques may not display uniform mortality following CHAPV infection, these animals recapitulate many key hemorrhagic and neurological clinical signs of CHF observed in the course of human disease. Thus, they are well-suited for evaluating the efficacy of certain antiviral candidates and for investigating the virus’s pathogenesis and persistence.

### 5.3. Summary of CHAPV Animal Models

While few published studies exist, cynomolgus macaques and Strain 13 guinea pigs both show promise as CHAPV animal models, albeit with important limitations. Cynomolgus macaques can develop hemorrhagic and neurologic signs (e.g., hypothermia, epistaxis, tremors), making them valuable for exploring CHF disease manifestations and virus persistence over time. However, infection in macaques did not lead to high mortality, which can complicate therapeutic testing that benefits from a uniformly lethal model. In contrast, Strain 13 guinea pigs uniformly succumb to CHAPV infection, offering an efficient platform to assess vaccine or antiviral efficacy. Yet, these animals largely failed to display robust neurological clinical signs. Additionally, the single-sex differences seen in guinea pigs (with female animals showing delayed progression) warrant further investigation to ensure reproducibility and a thorough understanding of CHAPV pathogenesis. Given the small sample sizes of animals tested in either case, more work is needed to refine these models and more fully characterize their potential for use in pathogenesis studies and preclinical MCM development.

## 6. Sabiá Virus (SABV)

### 6.1. Background

The first case of Sabiá virus was reported in 1990, when a woman who lived in São Paulo state, Brazil, became infected [124]. She was admitted to a hospital following a 12-day-long period of fever, headache, muscle aches, nausea, vomiting, and general weakness, where she stayed for 4 days until her death [124]. Her condition worsened over her time in hospital, with hematemesis, vaginal bleeding, conjunctival petechiae, somnolence, tremors, walking difficulties, and tonic-clonic convulsions all observed [124]. She went into a coma 3 days post-hospitalization/15 days post-symptom onset, resulting in her death the subsequent day [124]. A blood sample was obtained from this case, from which a then-unknown agent (now known as Sabiá virus, an arenavirus in the Tacaribe complex) was isolated and subsequently characterized through complement-fixation, immunofluorescence, and neutralization [124]. During characterization, a lab technician was infected through suspected aerosol exposure in his lab [124]. Like the index case, he presented with non-specific febrile symptoms and vomiting, but exhibited unique symptoms including bleeding gums, chills, malaise, stomach pain, and gingival hemorrhages [124]. After 15 days, the individual recovered [124]. Since these two cases occurred, there have only been four other cases of SABV infection, one of which was laboratory-acquired (also via aerosolized exposure), while the remainder were acquired through environmental exposure [125,126,127]. The other laboratory-acquired case resulted in the survival of the patient after a 13-day-long illness [127]. Notably, the lab-acquired infection patient was treated with ribavirin intravenously, which has been used to treat other NWAV infections [29,127]. The three other cases occurred in 1999, 2019, and 2020, respectively, in varying localities within the state of São Paulo [125,126]. All cases presented as a non-specific febrile illness, as earlier cases had, with the latter two demonstrating some neurological symptoms (namely confusion, stupor, and/or somnolence) [125,126]. Death was reported for the 1999 case after seven days of hospitalization, 12 days from the onset of symptoms for the 2019 case, and 13 days after the onset of symptoms for the 2020 case [125,126]. All cases were transmitted through environmental exposure or in laboratory settings; no human–human transmission has been observed for SABV [126,127].

### 6.2. Animal Models of Experimental Infection with Sabiá Virus

No animal models for SABV infection have been described to date.

## 7. Conclusions

This review of New World arenavirus animal models highlights several key findings. Effective models range from rodents—such as guinea pigs, rats, mice (in both immunocompetent and immunocompromised forms), and hamsters—to non-human primates, including rhesus macaques, cynomolgus macaques, African green monkeys, marmosets, capuchin monkeys, and Geoffroy’s tamarins. Among these, guinea pigs, mice, rhesus macaques, and cynomolgus macaques are the best models due to their ability to closely replicate human disease pathogenesis/clinical manifestations and/or their near-uniform mortality following infection. These traits make them ideal for pathogenesis studies and the development of MCMs.

For specific viruses, the best-suited models are as follows: guinea pigs and cynomolgus macaques for GTOV; Strain 13 guinea pigs cynomolgus macaques for CHAPV; rhesus macaques, guinea pigs, AGMs, and certain strains of mice (both immunocompromised and immunocompetent) for MACV; and rhesus macaques, guinea pigs, marmosets, certain rat strains (Wistar and Buffalo/Sims), and mice for JUNV. Non-human primates generally provide better models for recapitulating the clinical manifestations of NWAVs, including hemorrhagic, systemic, and neurological clinical signs, often with high mortality. However, their use is limited by high costs and logistical challenges.

Certain rodent models, like guinea pigs, offer uniform mortality and can partially replicate human disease pathologies. However, limited immunological reagents are available for guinea pigs. Mice, especially when immunocompromised or infected at a very young age, are effective for modeling neurological complications, but not hemorrhagic clinical signs, despite being more cost-effective than non-human primates. Moreover, immunocompromised mice may not be suitable for immunological studies. The effectiveness of a specific animal model varies with several factors, such as virus type, strain, inoculum size, animal age, and infection route. These factors should be carefully considered in experimental design to maximize research utility.

Given the ongoing public health threat posed by NWAVs and the lack of viable MCMs, animal models remain crucial for understanding disease pathogenesis and testing potential vaccines and antivirals. Future research should focus on developing models for under-characterized viruses like CHAPV, SABV, and GTOV, with guinea pigs being a promising candidate for further study. For better-characterized viruses, existing models will continue to play a vital role in evaluating new MCMs.

## Figures and Tables

**Figure 1 microorganisms-13-01358-f001:**
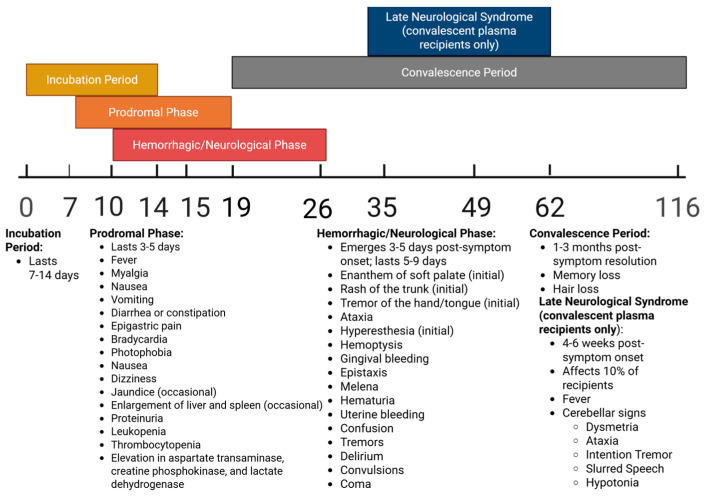
Timeline of stages and associated clinical manifestations of JUNV infection in humans, measured in days post-infection.

**Figure 2 microorganisms-13-01358-f002:**
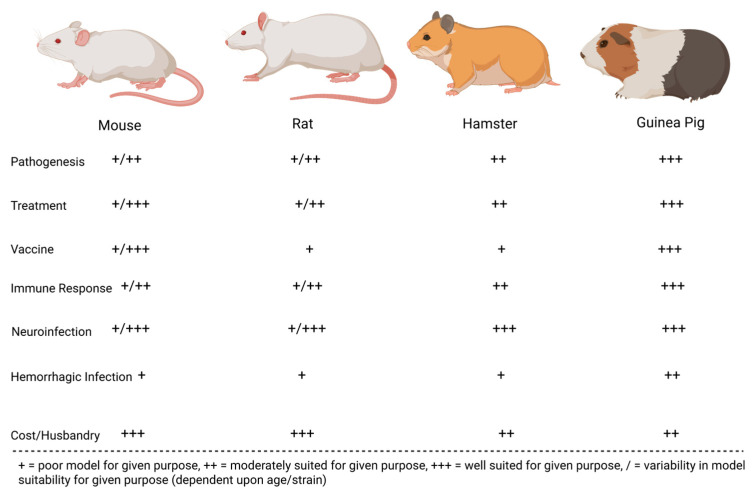
JUNV rodent models evaluated on their immune response to infection, economic viability, accuracy in recapitulating human infection (neurological or hemorrhagic), and their suitability for vaccination, treatment, and pathogenesis studies.

**Figure 3 microorganisms-13-01358-f003:**
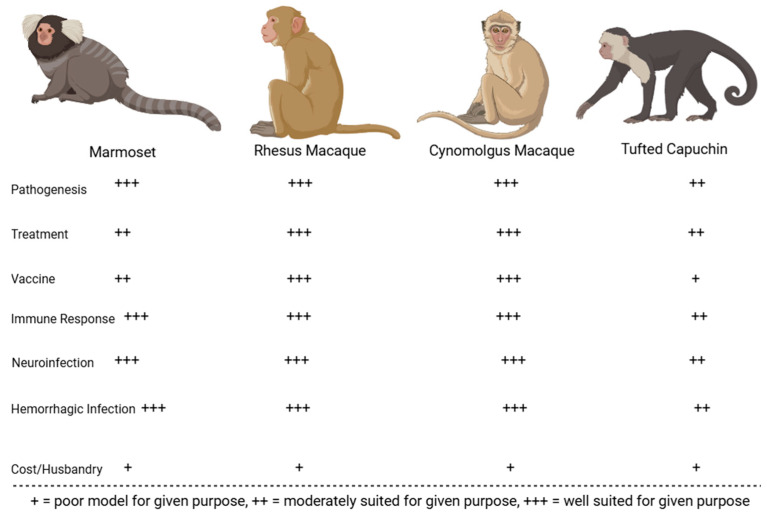
JUNV NHP models evaluated on their immune response to infection, economic viability, accuracy in recapitulating human symptoms of infection with virus (neurological or hemorrhagic), and suitability for vaccination, treatment, and pathogenesis studies.

**Figure 4 microorganisms-13-01358-f004:**
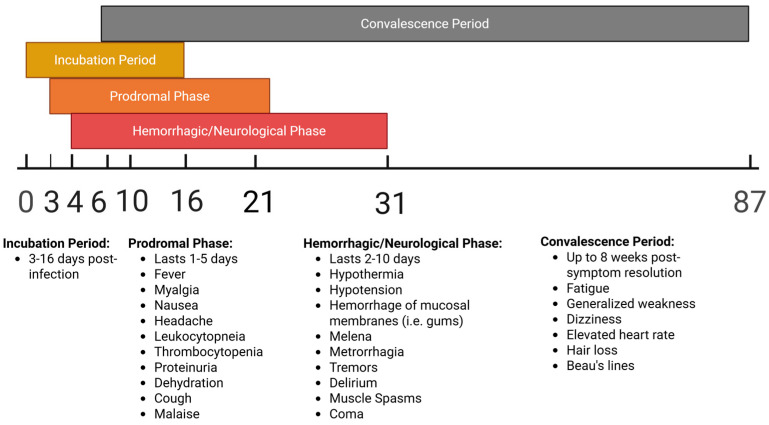
Timeline of stages and associated clinical manifestations of MACV infection in humans, measured in days post-infection.

**Figure 5 microorganisms-13-01358-f005:**
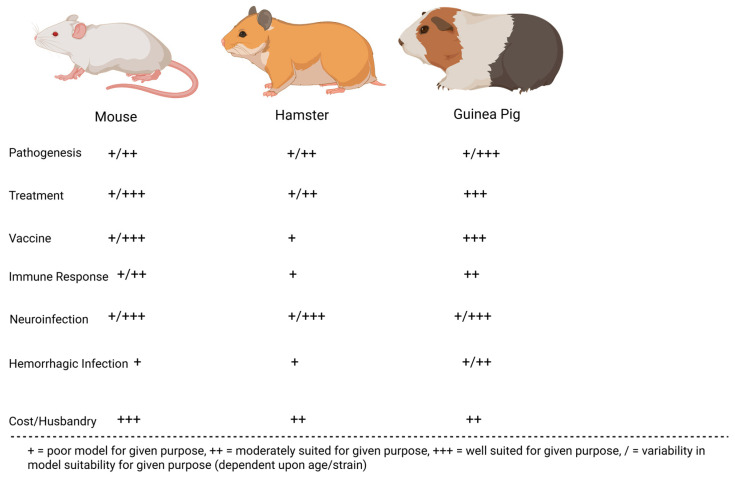
MACV rodent animal models evaluated on their immune response to infection, economic viability, accuracy in recapitulating human symptoms of infection with virus (neurological or hemorrhagic in nature), and suitability for vaccination, treatment, and pathogenesis studies.

**Figure 6 microorganisms-13-01358-f006:**
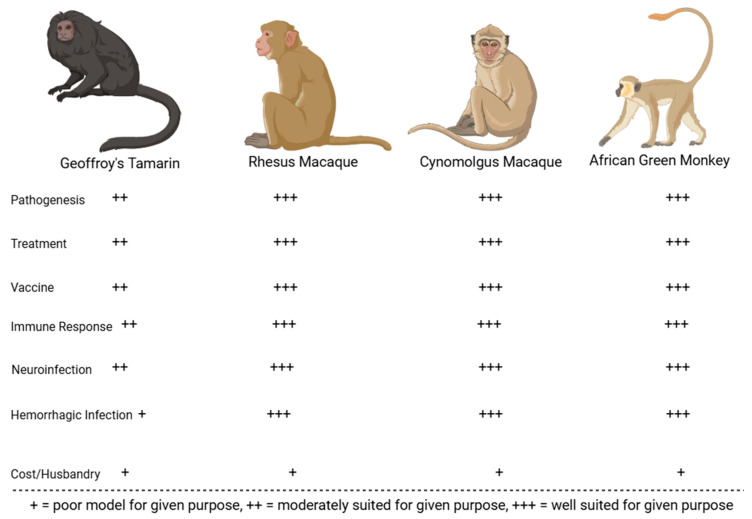
MACV NHP animal models evaluated on their immune response to infection, economic viability, accuracy in recapitulating human symptoms of infection with virus (neurological or hemorrhagic in nature), and suitability for vaccination, treatment, and pathogenesis studies.

**Figure 7 microorganisms-13-01358-f007:**
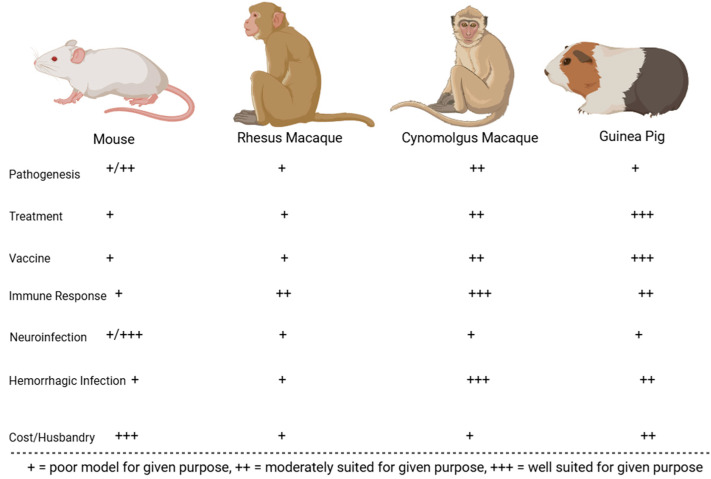
GTOV animal models evaluated on their immune response to infection, economic viability, accuracy in recapitulating human symptoms of infection with virus (neurological or hemorrhagic in nature), and suitability for vaccination, treatment, and pathogenesis studies.

## Data Availability

No new data were generated. All data supporting reported results can be found within the manuscript or linked references.

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
