# Peer review of "Animal Models of Pathogenic New World Arenaviruses"

_microorganisms, 2025, doi:10.3390/microorganisms13061358_

Round 1
Reviewer 1 Report
Comments and Suggestions for Authors
As a whole, the manuscript is well written and contains informative details regarding the state of animal models in the field. However, presentation of the material can be re-organized for clarity.
Introduction: This is generally well written, however, the flow of information is choppy and can be improved. There is a jump from general epidemiology info to genetics to host cell entry to countermeasures. Since the focus of this review is animal models, it may be better for the readers to remove the individual sections later on about epidemiology and clinical manifestations and keep that generalized to the introduction. A table or figure on clinical symptom onset could be beneficial in this section
Figures are repetitive. It would be more appropriate to have one figure of each style (rodent and NHP) that addresses the viruses in the family that are appropriate for each subtopic (pathogenesis, treatment, etc) instead of replicating near-identical figures for each virus subsection.
While the tables are extensive and information, they may be better suited as written as supplemental tables and shorter, more informative tables can be inserted within the text. For example, in Table 1, there are rodents of the same strain at various ages as their own line item but the details of virus strain, clinical disease, etc are very similar. These could be combined.
Reviewer 2 Report
Comments and Suggestions for Authors
For the five types of sarcovirus that pose a continuous and serious threat to human public health (with a case fatality rate basically above 30%) and lack specific medical countermeasures (lacking safe and effective vaccines and therapeutic drugs), the morbidity, clinical symptoms and lesions caused by these five pathogens in various parts of the world over a long period of time are described in detail. The pathogenic model animals of four types of sarcovirus were introduced in particular detail, providing model tools for future research on the pathogenic mechanism, clinical manifestations, pathological changes, disease prognosis of these diseases, as well as the development and screening of safe and effective vaccines and therapeutic drugs. This is conducive to in-depth research and correct diagnosis and differentiation of these diseases. It will play a promoting role in the development of safe and effective vaccines and therapeutic drugs for these diseases.
Several diseases caused by the sand virus pathogen are placed in one article, allowing people to systematically and comparatively understand and identify these diseases and their model animals.
Overall, the author failed to summarize, sort out and process the literature materials well. Instead, they listed the materials in a general way. For example: (1) A table can be used to sort out the commonalities and individualities of the clinical symptoms and pathological changes of the five diseases. (2) Figures 1 to 5 are summarized into one figure; (3) Tables 1 to 16 should be transformed into four tables, or simply summarized in words instead of listing them one by one. It is absolutely not allowed to describe the daily performance after the drug attack and so on.
Errors: (1) The textual description part after Table 6 was written as Table 8, and the textual description part after Table 8 was written as Table 6. (2) In many places in the text, only the last digit of two starting and ending percentages is followed by a percentage sign. Such as 15%-30% instead of 15%-30%.
